# Critical roles of metal–ligand complexes in the controlled synthesis of various metal nanoclusters

Ji Soo Kim [1,2], Hogeun Chang [1,2,5], Sungsu Kang[1,2], Seungwoo Cha[2,6], Hanguk Cho [1,2], Seung Jae Kwak[2], Namjun Park [2], Younhwa Kim[1,2], Dohun Kang [1,2,7], Chyan Kyung Song[1,2], Jimin Kwag[1,2], Ji-Sook Hahn [2], Won Bo Lee [2], Taeghwan Hyeon [1,2,8] ✉ & Jungwon Park [1,2,3,4,8] ✉

Metal nanoclusters (NCs), an important class of nanoparticles (NPs), are extremely small in size and possess quasi-molecular properties. Due to accurate stoichiometry of constituent atoms and ligands, NCs have strong structure-property relationship. The synthesis of NCs is seemingly similar to that of NPs as both are formed by colloidal phase transitions. However, they are considerably different because of metal-ligand complexes in NC synthesis. Reactive ligands can convert metal salts to complexes, actual precursors to metal NCs. During the complex formation, various metal species occur, having different reactivity and fraction depending on synthetic conditions. It can alter their degree of participation in NC synthesis and the homogeneity of final products. Herein, we investigate the effects of complex formation on the entire NC synthesis. By controlling the fraction of various Au species showing different reactivity, we find that the extent of complex formation alters reduction kinetics and the uniformity of Au NCs. We demonstrate that this concept can be universally applied to synthesize Ag, Pt, Pd, and Rh NCs.

Synthesis of highly monodisperse nanoparticles (NPs) is of primary importance in various fields as their physical properties strongly depend on their atomic structure[1–3]. Based on this perspective, metal nanoclusters (NCs) have emerged as a unique class of NPs which are defined by molecular formula[4,5]. They are composed of a precise number of metal atoms and ligands, ranging from less than ten to a few hundred. Such atomic precision and resultant ultrasmall size endow metal NCs with discrete energy levels, similar to molecular electronic structures[6]. During the metal NC synthesis, a phase transition occurs from the solution containing metal precursors to solid NCs formed

through metallic bonding[7,8]. This formation process seemingly resembles the conventional colloidal synthesis of inorganic NPs. NP formation is explained by nucleation whereby monomers assemble into solid-phase particulates, and their growth occurring via several routes[9–11]. All steps in the NP synthesis are governed by the chemical potential of monomers and surface energy of the growing NPs. Upon supersaturation of monomers, a burst of nucleation occurs, after which small particulates grow to large size to decrease free energy[12,13]. However, this description is not readily applicable to NC formation. There are metal–ligand complexes, formed by the interactions

[1]Center for Nanoparticle Research, Institute for Basic Science (IBS), Seoul National University, Seoul 08826, Republic of Korea. [2]School of Chemical and Biological Engineering, and Institute of Chemical Process, Seoul National University, Seoul 08826, Republic of Korea. [3]Institute of Engineering Research, College of Engineering, Seoul National University, Seoul 08826, Republic of Korea. [4]Advanced Institute of Convergence Technology, Seoul National University, Suwon-si, Gyeonggi-do 16229, Republic of Korea. [5]Present address: Samsung Advanced Institute of Technology, Samsung Electronics, Suwon 16678, Republic of Korea. [6]Present address: Bio-MAX/N-Bio, Institute of BioEngineering, Seoul National University, Seoul, Republic of Korea. [7]Present address: Department of Materials Science and Engineering, Northwestern University, Evanston, IL 60208, USA. [8]These authors jointly supervised this work: Taeghwan Hyeon, Jungwon Park. ✉e-mail: thyeon@snu.ac.kr; jungwonpark@snu.ac.kr

between metal precursors and reactive ligands[14–16]. Those complexes are produced as intermediate species and subsequently act as actual precursors to metal NCs, the formation of which is driven by external stimuli[17,18].

Among various atomically precise metal NCs, thiolate-protected metal NCs are representative examples[19–22]. The strong affinity of sulfur to metal readily converts metal salts to metal–thiolate complexes, and then they are transformed into metal NCs by reducing agents[23,24]. Before the formation of metal NCs, there exist various types of complexes with respect to the number of metal atoms and thiol ligands[25,26]. In addition, the strong metal–sulfur bond renders the complexes sufficiently stable for fine control of their reduction kinetics, allowing the successful synthesis of metal NCs[27]. The various metal species, emerging from reaction between metal salt and thiol ligands, can exhibit dissimilar reactivity toward reducing agents owing to different coordination of metal atoms[28,29]. Moreover, the fraction of each metal species varies sensitively depending on the reaction condition. This can eventually alter their degree of participation as precursors in metal NC synthesis. Thus, to elucidate the formation mechanism of metal NCs and control their ensemble uniformity, it is crucial to understand how diverse metal species with different reactivity are involved in the overall NC formation process.

Herein, we investigate the synthetic principle of thiolate-protected, water-soluble metal NCs and elucidate the importance of controlling metal–thiolate complex formation. Au NC synthesis is chosen as a representative system, where both complexes and NCs are controlled under different synthetic conditions and characterized using various techniques. We reveal that the extent of reaction between Au salts and thiol ligands governs the fraction of Au–thiolate complexes showing different reactivity, which directly controls both the reduction kinetics and the homogeneity of the final Au NCs. Furthermore, based on this understanding, we demonstrate that tuning metal–ligand complex formation universally facilitates the controlled synthesis of thiolate-protected metal NCs composed of Ag, Pt, Pd, or Rh.

## Results

### Effect of the complex formation on the homogeneity of Au NC

Thiolate-protected Au NCs are representative metal NCs which undergo complex-mediated formation[19,30,31]. They are synthesized through two consecutive reduction process: from $Au^{III}$ salts to $Au^{I}$-SR complexes by thiol ligands (the first reduction), and from Au-SRs to Au NCs by reducing agents (the second reduction)[23,32]. We synthesized water-soluble $Au_{25}(MHA)_{18}$ (6-mercaptohexanoic acid) NCs via such a two-step reduction process[33]. In the first reduction, $HAu^{III}Cl_4$ directly reacts with MHA ligands to form Au–MHA complexes under basic conditions. Those complexes are subsequently reduced using $NaBH_4$ to form $Au_{25}$ NCs. The extent of the complex formation was controlled by changing two major factors: the duration of the first reduction and the ratio of Au to MHA (Supplementary Figs. 1,2). The other experimental parameters were kept constant in all syntheses, particularly, the second reduction time of 3 h and the amount of $NaBH_4$. UV-vis absorption spectra, and matrix-assisted laser desorption/ionization time-of-flight (MALDI-TOF) mass spectra were utilized to characterize Au NCs to assess their homogeneity. Under all conditions, atomically precise $Au_{25}$ NCs are successfully synthesized, regardless of changes in the first reduction. However, we find that the homogeneity of Au NCs varies sensitively according to the degree of the first reduction. A longer first reduction time or a more amount of MHA produces more homogeneous $Au_{25}$ NCs without the formation of additional NCs. When the first reduction time and amount of MHA are decreased, large Au NCs are synthesized concurrently with $Au_{25}$ NCs, indicating a decline in homogeneity. Other than the two parameters, there exist other parameters, reactant concentration or temperature, which can control the Au NC synthesis[34,35]. We examined how they affect the first

reduction or homogeneity of produced Au NCs (Supplementary Figs. 3,4). Concentration or temperature shows less effect on both the degree of first reduction and Au NC homogeneity than the first reduction time and the ratio of Au precursor to thiol ligands (Methods).

The Au NCs obtained via the four representative systems from Supplementary Figs. 1, 2, two for each of the two scenarios, were characterized (Fig. 1). Using a small amount of MHA and a short first reduction time (Au: MHA = 1:1, 15 s and Au: MHA = 1:2, 15 s) yields Au NCs with decreased homogeneity. In contrast, uniform $Au_{25}$ NCs are synthesized using a large amount of MHA or a long first reduction time (Au: MHA = 1:3, 15 s and Au: MHA = 1:2, 7 d). As the homogeneity of Au NCs decreases, distinct $Au_{25}$ NCs absorption peaks, centered at approximately 400 nm, 450 nm, and 670 nm in UV-vis spectrum (dashed lines in Fig. 1a)[33], become less prominent, whereas the uniform Au NCs achieve high purity of $Au_{25}$ NCs as confirmed by their optical density at 670 nm (Methods). Furthermore, the existence of NCs larger than $Au_{25}$ NCs is verified by an additional peak in the MALDI-TOF mass spectrum of the less homogeneous Au NCs (arrows in Fig. 1b), whereas it disappears in the other two cases of more homogeneous $Au_{25}$ NCs. Polyacrylamide gel electrophoresis (PAGE) was also employed to verify the differences in the homogeneity of the Au NCs obtained under each conditions (Supplementary Fig. 5). While the lanes of the less uniform Au NCs show several bands for Au NCs larger than $Au_{25}$ NCs, the other two lanes display dominant $Au_{25}$ NCs bands.

The differences in the homogeneity of Au NCs were further analyzed by extended X-ray absorption fine structure (EXAFS), obtained at the Au L3 edge (Fig. 1c). The position of Au-Au peaks was assigned by the EXAFS data for Au foil and Au NPs as references (Supplementary Fig. 6)[36]. The Au-Au peaks of the less homogeneous Au NCs are more intense than those of the more homogeneous $Au_{25}$ NCs. Considering that $Au_{25}(SR)_{18}$ NCs show suppressed Au-Au peaks in EXAFS at room temperature[37], it is evident that the first reduction with high extent produces more homogeneous $Au_{25}$ NCs, whereas the increased Au-Au peaks from less homogeneous Au NCs are attributed to the Au NCs larger than $Au_{25}$ NCs. Additionally, differences in the uniformity of the Au NCs are visible in the transmission electron microscopy (TEM) images (Fig. 1d–k and Supplementary Fig. 7). Aberration-corrected TEM ($C_s$-TEM) using graphene grids allows high resolution imaging of ultrasmall Au NCs[38]. Large Au NCs are observed in the two cases of the first reduction with low extent (Fig. 1d, e and Supplementary Fig. 7a, b), which results in an increase in the average size and standard deviation (Fig. 1h, i). In contrast, the formation of uniform Au NCs is confirmed for the opposing cases, as evidenced by the small average size and standard deviation (Fig. 1f, g,j,k and Supplementary Fig. 7c, d).

### Effect of first reduction on the fraction of Au–SR complexes

We hypothesize that the different homogeneity of Au NCs is the result of varying fraction of Au–thiolate complexes, which arise based on the degree of the first reduction. We characterized Au-MHA complexes under the four conditions shown in Fig. 1 before the second reduction (Fig. 2). All characterizations were performed after washing out the residual ligands and $Au^{III}$ salts to prevent undesired reaction. Differences between the Cl 2$p$ spectra of the unwashed and washed samples in X-ray photoelectron spectroscopy (XPS) verifies the removal of Au salt in each case (Supplementary Fig. 8). $Au_4(MHA)_4$ is detected as the major species along with other minor complexes (Fig. 2a). The accurate number of Au and MHA in those complexes is confirmed by the isotope calculation (Supplementary Fig. 9). The compositions of the detected complexes under each condition are seemingly similar. Additionally, those complexes in solution state form single-sized aggregation with similar hydrodynamic diameter investigated by dynamic light scattering (DLS), regardless of synthetic conditions (Supplementary Fig. 10)[34].

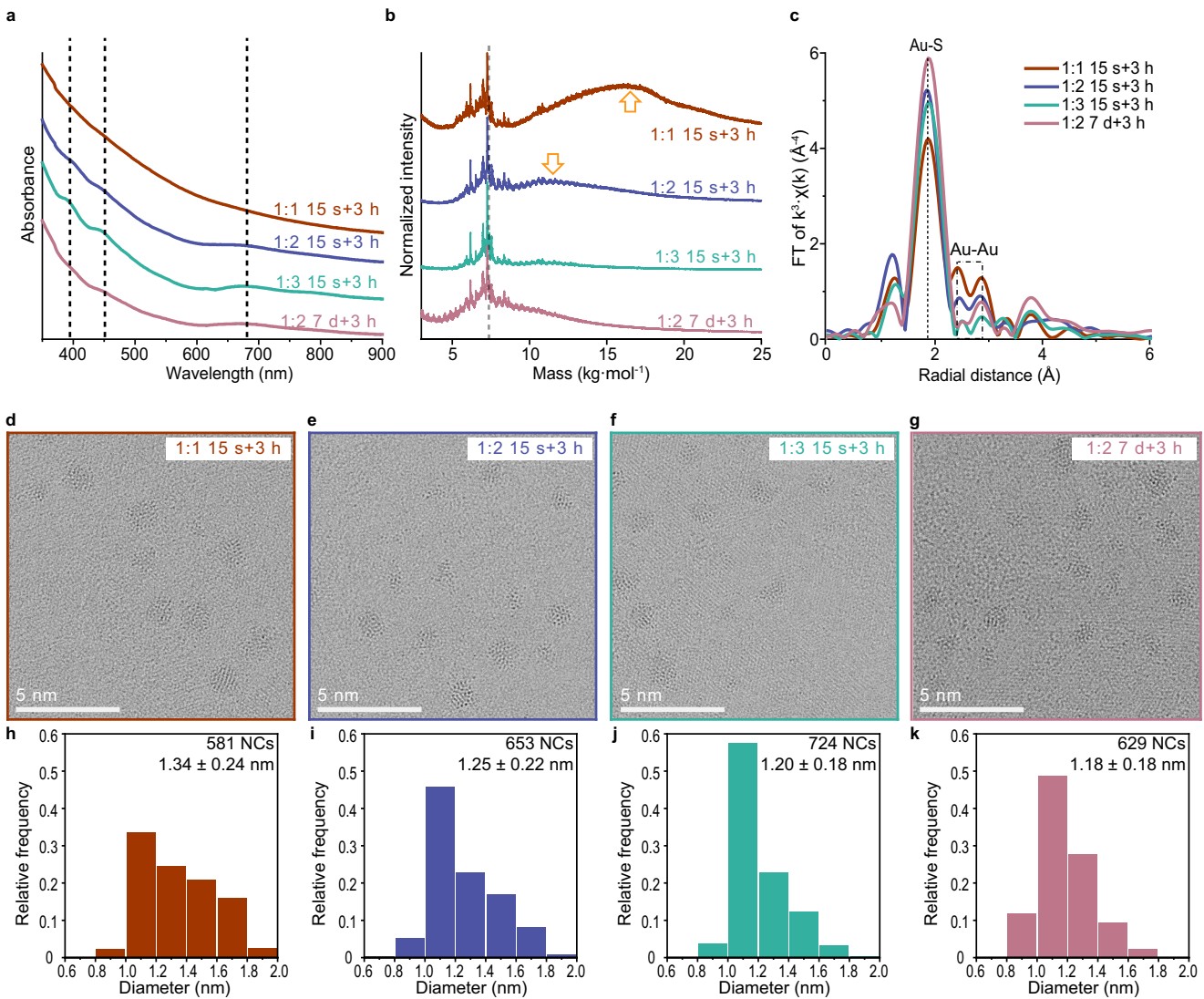

**Fig. 1 | Different homogeneity of Au NCs obtained under representative conditions. a** UV–vis absorption spectra. **b** MALDI-TOF mass spectra (in the linear mode) of the less and more homogeneous Au NCs, respectively (1:1, 15 s brown; 1:2, 15 s blue; 1:3, 15 s turquoise, 1:2, 7 d pink). Dashed lines in (**a**, **b**) indicate the evidence of $Au_{25}$ NCs. **c** EXAFS spectra of Au NCs under the indicated synthetic conditions. Dashed line indicates the Au-S peak, and dashed rectangle indicates the Au-Au peak. **d–g**, Representative $C_s$-TEM images of Au NCs under each conditions. **h–k**, Size distribution of the Au NCs from (**d–g**), respectively. Source data are provided as a Source Data file.

However, their amounts produced from Au salts differ depending on the degree of the first reduction. The yields of Au–MHA complexes under the four conditions were acquired by inductively coupled plasma atomic emission spectroscopy (ICP-AES) (Fig. 2b). The yield of complexes is positively correlated with the extent of the first reduction; the longer the first reduction time and the more amount of MHA is used, the higher yield of the complexes is obtained.

Not only the amount of Au–MHA complexes, difference in the first reduction also changes the fraction of the various complexes within themselves. This was confirmed by the changes in the ensemble oxidation state and coordination number of Au–S bonds in the complexes, as assessed by X-ray absorption near edge structure (XANES) and EXAFS (Fig. 2c–g). The absorption edge in the XANES spectra shifts to higher energy as the degree of the first reduction increases (Fig. 2c). Considering that residual $Au^{III}$ salts from the first reduction was already removed (Supplementary Fig. 8), such a difference in the edge energy of the spectra implies a dissimilarity in the coordination number of Au with S in $Au^{I}$–MHA complexes, which alters ensemble oxidation state of Au[39]. The differences in Au-S bonds were further analyzed by EXAFS

(Fig. 2d–g). Combined with the calculated parameters for Au-S bonds (Supplementary Figs. 11, 12 and Supplementary Table 1), the coordination number of these bonds in the complexes approaches 2 when the first reduction is prolonged or a high amount of MHA is used. It indicates that the fraction of the Au–MHA complexes, wherein all Au atoms are coordinated with two S such as the calculated structure of $Au_3(MHA)_3$, $Au_4(MHA)_4$, and $Au_5(MHA)_5$ (Fig. 2i-k)[25,40,41], increases with a more intense first reduction. Such complexes are more stable for fine control of the second reduction than those possessing Au with only one Au-S bond such as $Au_2(MHA)_2$ (Fig. 2h), which is also supported by the lower energies of $Au_{3-5}(MHA)_{3-5}$ than those of $Au_2(MHA)_2$ (Fig. 2l–n and Methods)[40,41]. It implies that the increased fraction of more stable Au–MHA complexes leads to the successful synthesis of homogeneous $Au_{25}$ NCs.

It is commonly accepted that the Au-SR complexes produced from the first reduction are the major precursors to $Au_{25}$ NCs. However, it is not surprising that the Au salts remaining after the first reduction, if any, are reactive for the second reduction[25,28]. The ICP-AES results shows that the conversion of the Au salts to Au–MHA complexes in the first reduction is not in unity yield, indicating the

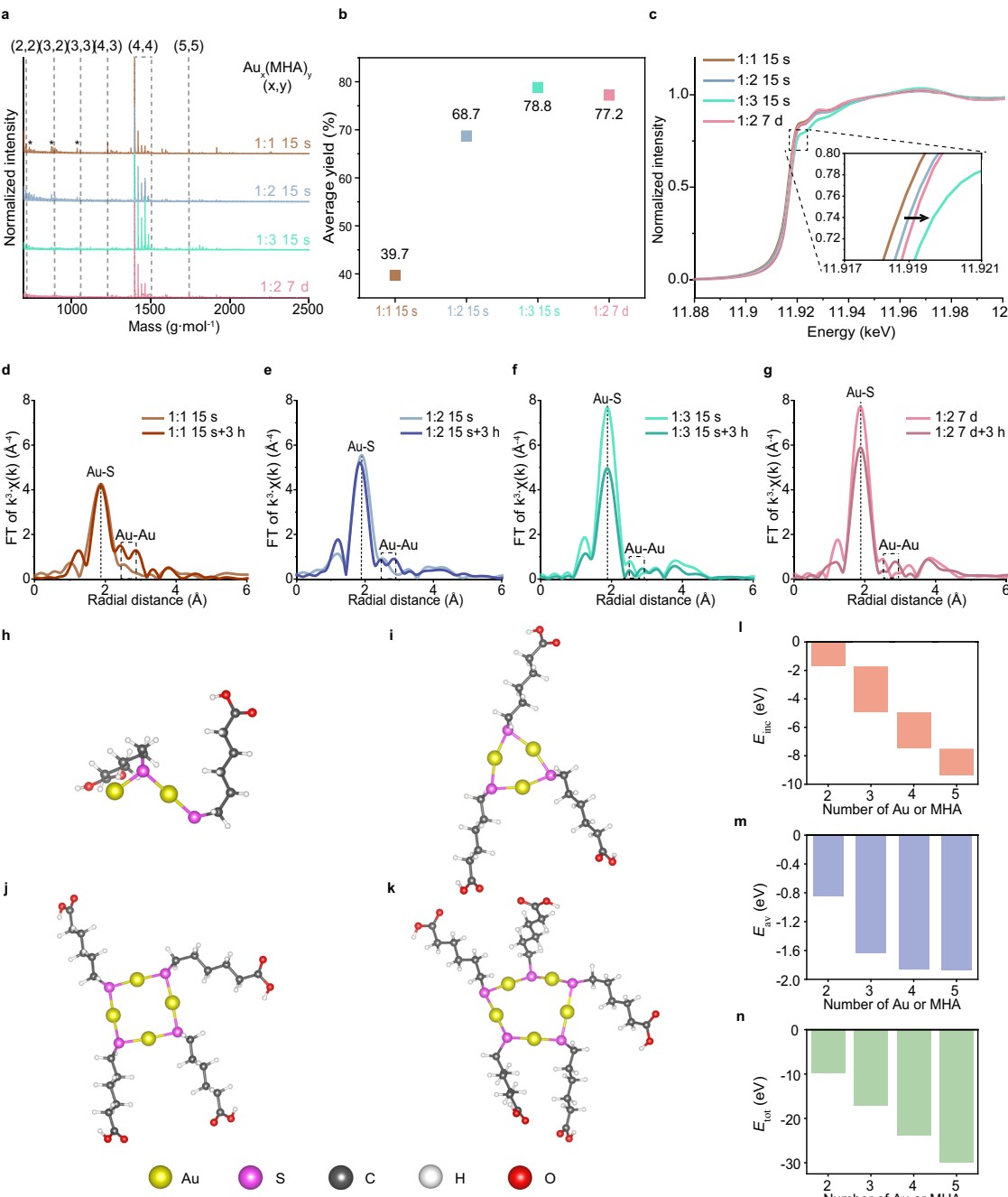

**Fig. 2 | Different fraction of Au–MHA complexes obtained under representative conditions. a** MALDI-TOF mass spectra (in the reflector mode) of Au-MHA precursors under different conditions (1:1, 15 s light brown; 1:2, 15 s light blue; 1:3, 15 s light turquoise, 1:2, 7 d light pink). Dashed lines indicate the Au–MHA complexes matched with the number of Au and MHA in parentheses, and dashed rectangles indicate the Au4(MHA)4 family with Na+ adducts. **b** Yield of the Au-MHA complexes under the indicated conditions obtained by ICP-AES. **c** XANES spectra of the Au–MHA complexes under the indicated conditions. Inset, shift in oxidation state of Au–MHA complexes. **d–g**, EXAFS spectra of Au-MHA complexes (pale color) and Au NC product (bold color) under the indicated synthetic conditions. Dashed line in each EXAFS spectra indicates the Au-S peak, and dashed rectangle indicates the Au-Au peak. **h–k** Calculated structures of Au2(MHA)2, Au3(MHA)3, Au4(MHA)4, and Au5(MHA)5, respectively (Au: yellow, S: magenta, C: dark gray, H: white, O: red). **l–n** Incremental fragmentation energy ($E_{inc}$, color: coral), average monomer fragmentation energy ($E_{av}$, color: azure), and overall dissociation energy ($E_{tot}$, color: olive) of the four complexes of (**h–k**). The energy of the Au(MHA) complex in (**m, n**) is set as 0. Source data are provided as a Source Data file.

existence of remaining Au salts before the second reduction (Fig. 2b). Furthermore, comparing EXAFS spectra of washed Au–MHA complexes and Au NCs, the intensity of Au-S peak at 1.87 Å remains almost constant in the two cases of producing less homogeneous Au NCs (Fig. 2d,e). It implies that there are additional types of precursors involved in Au NCs formation other than Au–MHA complexes. In contrast, the intensity of Au-S peak at 1.87 Å considerably decreases in

the other two cases (Fig. 2f,g), implying that the use of Au–MHA complexes enhances the homogeneity of Au25 NCs in the second reduction. In addition, the solution of Au NCs synthesized under the condition of Au: MHA = 1:1 and 15 s of first reduction time does not contain [Au(MHA)2]− complex, the well-known byproduct from the reduction reaction of Au-SR complexes to Au25 NCs (Supplementary Fig. 13)[26]. It infers that, when the extent of complex formation is low, Au

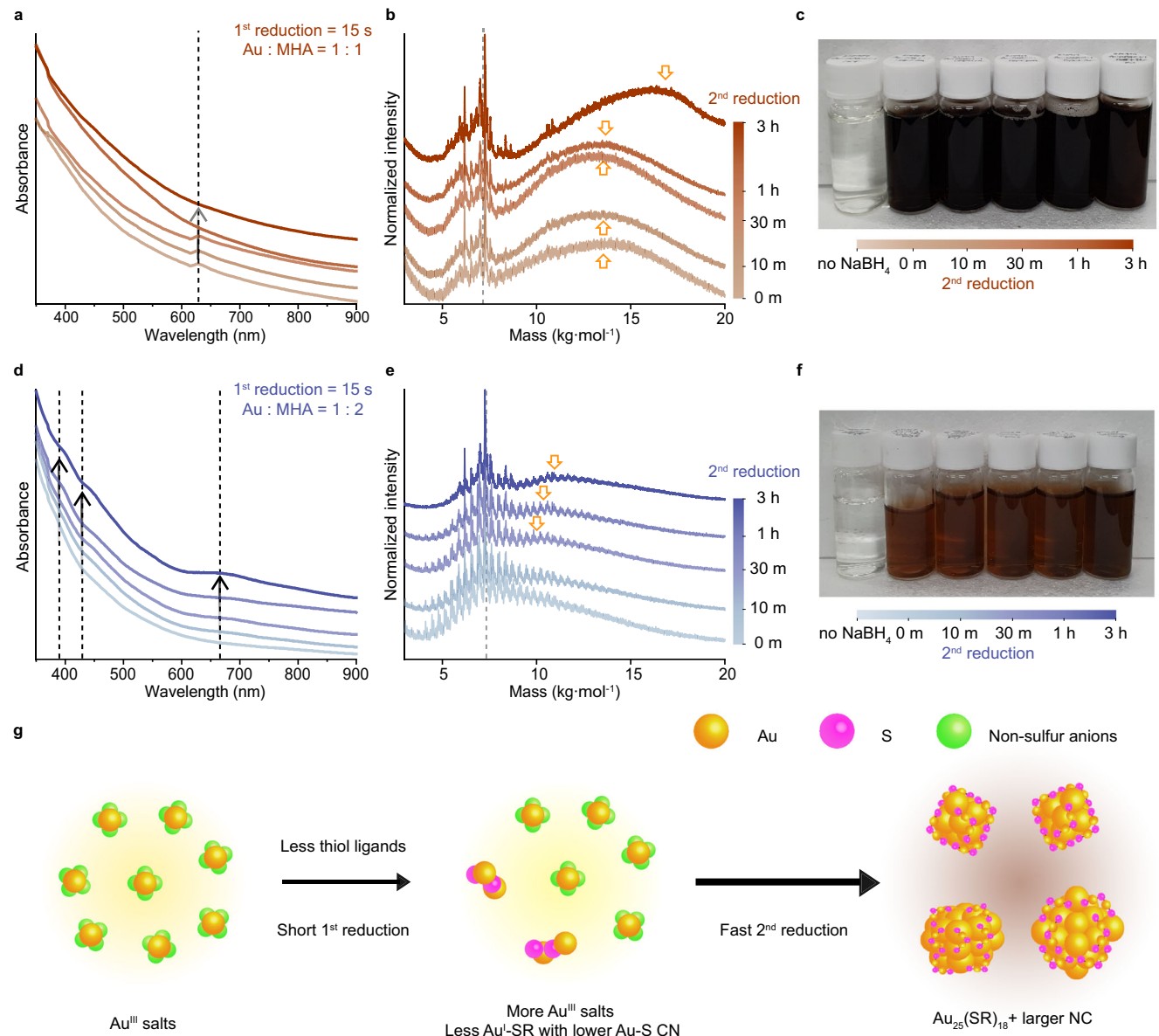

**Fig. 3 | Fast second reduction kinetics and less homogeneous Au NCs.**
**a**, **d** UV–vis absorption spectra, **b**, **e** MALDI-TOF mass spectra in the linear mode, and (**c**, **f**), Digital photographs of Au NCs during the second reduction using precursors obtained applying the 1:1, 15 s (from 0 m of second reduction, light brown to 3 h of second reduction, bold brown), and the 1:2, 15 s conditions (from 0 m of second reduction, light blue to 3 h of second reduction, bold blue), respectively. Dashed lines in UV–vis absorption and mass spectra indicate the evidence of $Au_{25}$ NCs. Arrows in (**a**) indicate the decreasing fraction of $Au_{25}$ NCs with increasing second reduction time, while those in (**d**) indicate the increasing fraction of $Au_{25}$ NCs with increasing second reduction time. Arrows in (**b**, **e**) indicate the existence of Au NCs larger than $Au_{25}$. **g** Reaction mechanism scheme when the degree of the first reduction is low (Au: yellow, S: magenta, non-sulfur anions: lime). Source data are provided as a Source Data file.

NCs are synthesized from different synthetic route where the starting precursors are different from Au–SR complexes.

## Effect of coordination of Au species on reduction kinetics

The coordination of Au with different anion induces the Au precursors to show different reactivity toward reduction by $NaBH_4$. It alters the kinetics of the second reduction and eventually affects the size-dispersity of Au NCs[23]. We characterized sequentially obtained samples at specific time intervals during each second reduction under the four conditions shown in Fig. 1 (Figs. 3, 4). In the case of the less homogeneous synthesis, 1:1 Au to MHA ratio and 15 s of first reduction time (Fig. 3a–c), a weak peak near 650 nm in UV-vis spectra is noticeable until 30 min of the second reduction time but disappears after 1 h (Fig. 3a). It is attributed to the increasing fraction of large NCs as the

second reduction progresses (Fig. 3b). In the other case of less homogeneous Au NCs, 1:2 Au to MHA ratio and 15 s of first reduction time (Fig. 3d–f), the UV-vis absorption peaks of $Au_{25}$ NCs gradually become apparent with increasing second reduction time (Fig. 3d). The intermediate Au NCs are concurrently formed with $Au_{25}$ NCs after the initiation of second reduction[26], but their fraction decreases along with the reduction time (Fig. 3e and Supplementary Fig. 14). This trend is due to the focusing process of size-dispersed Au NCs by the enhanced etching ability of thiol ligands under basic condition[33]. Nonetheless, the homogeneity of the final product in this condition is not optimal because of the existence of NCs larger than $Au_{25}$ NCs. The mass of the larger NC is matched with $Au_{38}(MHA)_{24}$ NCs which appear when the amount of $NaBH_4$ exceeds the 'stoichiometric amount'[26]. In addition, the formation of Au NCs is rapidly initiated immediately following the

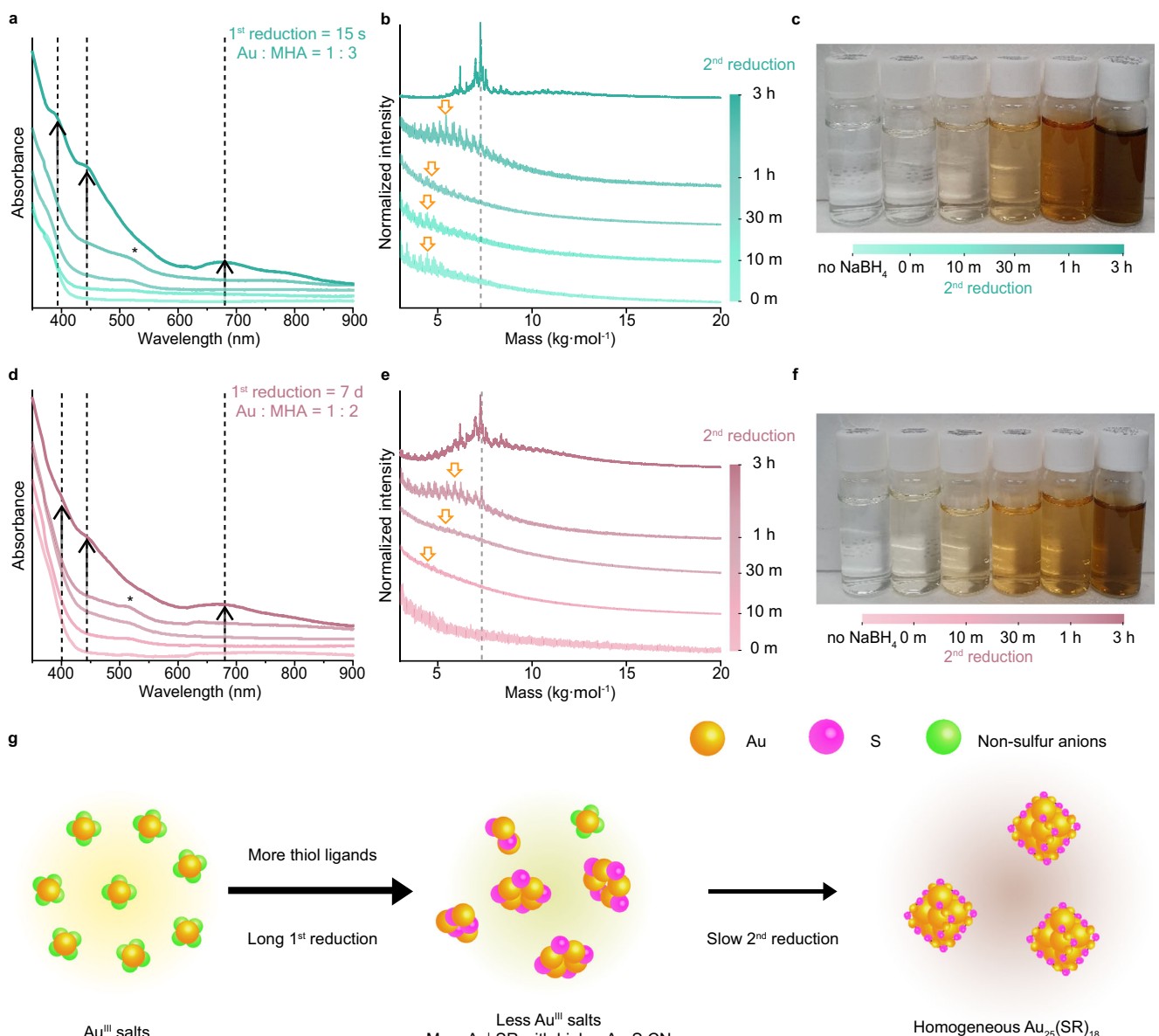

**Fig. 4 | Slow second reduction kinetics and more homogeneous Au NCs.**
**a, d** UV–vis absorption spectra, **b, e** MALDI-TOF mass spectra in the linear mode, and **c, f** Digital photographs of Au NCs during the second reduction using precursors obtained applying 1:3, 15 s (from 0 m of second reduction, light turquoise to 3 h of second reduction, bold turquoise) and 1:2, 7 d conditions (from 0 m of second reduction, light pink to 3 h of second reduction, bold pink), respectively. Dashed lines in UV–vis absorption and mass spectra indicate the evidence of Au$_{25}$ NCs. Arrows in (**a, d**) indicate an increasing fraction of Au$_{25}$ with increasing second reduction time. Arrows in (**b, e**) indicate the existence of Au NCs smaller than Au$_{25}$. Evidence of Au NCs smaller than Au$_{25}$ also appears in UV–vis absorption spectra (black asterisks). **g** Reaction mechanism schematic when the degree of the first reduction is high (Au: yellow, S: magenta, non-sulfur anions: lime). Source data are provided as a Source Data file.

NaBH$_4$ injection, as indicated by abrupt color change under both conditions (Fig. 3c, f). In summary, a schematic of the synthetic mechanism of less homogeneous Au NCs is proposed in Fig. 3g. Low degree of first reduction yields a small fraction of Au–MHA complexes, and the remaining Au salts majorly participate in the formation of Au NCs. The high reactivity of Au salts and the Au–MHA complexes, with relatively low Au-S coordination, induces fast second reduction kinetics and results in the synthesis of Au NCs with reduced homogeneity.

On the contrary, in the synthesis of more homogeneous Au$_{25}$ NCs, 1:3 Au to MHA ratio and 15 s of first reduction time (Fig. 4a–c), prominent UV–vis peaks of Au$_{25}$ NCs in the final product confirm the successful synthesis with enhanced homogeneity (Fig. 4a). Notably, a peak located at approximately 520 nm, presumably

corresponding to intermediate Au$_{18}$(MHA)$_{14}$ NCs[42–44], disappears after 1 h of second reduction time. It indicates that homogeneous Au$_{25}$ NC synthesis proceeds with the consumption of smaller intermediate NCs[25,26], also confirmed by mass spectra (Fig. 4b and Supplementary Fig. 15). This trend is also consistent in the synthesis with 1:2 Au to ligand ratio and 7 d of first reduction time, as evidenced by UV–vis spectra and mass spectra, obtained during the second reduction (Fig. 4d, e). The formation of homogeneous Au$_{25}$ NCs occurs with slow second reduction kinetics, as shown by gradual color change under both conditions (Fig. 4c, f). To summarize the synthetic mechanism as displayed in Fig. 4g, high degree of the first reduction consumes most of the Au salts, producing Au–MHA complexes with high Au-S coordination as the major species. The stability of these complexes enables the controlled synthesis of

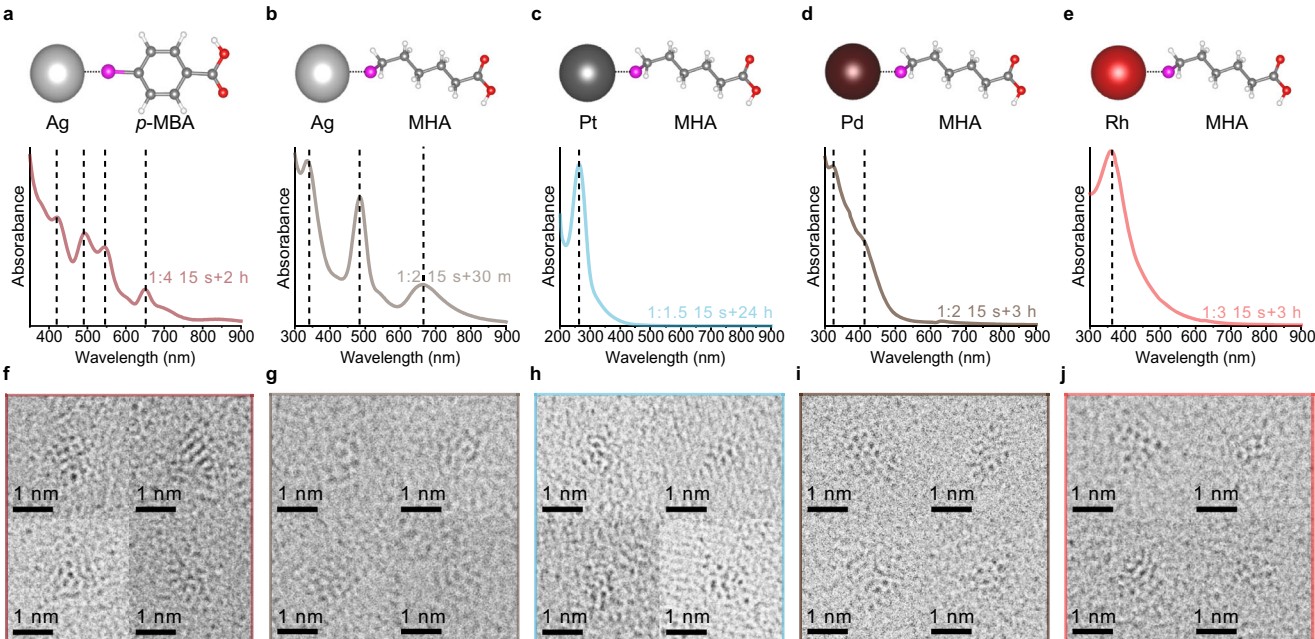

**Fig. 5 | Optimal conditions for synthesizing uniform NCs composed of Ag, Pt, Pd, or Rh. a–e** UV–vis absorption spectra of homogeneous (*p*-MBA)-protected Ag NCs (color: burgundy), MHA-protected Ag NCs (color: light gray), MHA-protected Pt (color: sky blue), Pd (color: auburn), and Rh NCs (color: light red), respectively. Dashed lines in **a**, **b** indicate the evidence of Ag$_{44}$(*p*-MBA)$_{30}$ NCs and Ag$_{25}$(MHA)$_{18}$ NCs, respectively. Dashed lines in (**c**–**e**) indicate the evidence of quasi-molecular properties of ultrasmall Pt, Pd, and Rh NPs, respectively. Schematic representations of metal–thiol interaction are described above the UV-vis absorption spectra. The colors of all atoms except for those of metal are identical with those in Fig. 2f-j (Ag: silver, Pt: gray, Pd: auburn, Rh: red). **f–j** Representative C$_s$-TEM images of homogeneous (*p*-MBA)-protected Ag NCs, MHA-protected Ag NCs, MHA-protected Pt, Pd, and Rh NCs, respectively. Source data are provided as a Source Data file.

homogeneous Au$_{25}$ NCs driven by slow second reduction kinetics. During the slow second reduction, according to recent reports, the Au(0) core of intermediate Au NCs is initially nucleated from the interaction between high fraction of Au$_n$(MHA)$_n$ complexes and BH$_4^-$ ions[45], and those intermediate NCs grow to larger ones via atom-by-atom size evolution process[8,46].

These results indicate that the coordination chemistry of Au species alters their reactivity and determines the kinetics of the second reduction. To confirm it, we compared the reduction rates of HAu$^{III}$Cl$_4$, Au$^{III}$(OH)$_3$, Au$^I$Cl, and Au$^I$-thiomalate using NaBH$_4$ as shown in Supplementary Fig. 16, 17. The two Au precursors coordinated with Cl exhibit extremely rapid reduction rates (Supplementary Fig. 16a, b). Au$^{III}$(OH)$_3$, which can be produced by the exchange of Cl from HAuCl$_4$ in basic media[47], is also abruptly reduced by NaBH$_4$ (Supplementary Fig. 16c). However, Au-thiomalate is reduced with relatively slow kinetics, implying that Au-S bonds provide stability to the Au–SR complex (Supplementary Fig. 17). Moreover, the washed Au-MHA precursors in the two cases of Fig. 3 undergo a drastically decelerated reduction by NaBH$_4$, compared to the original synthesis, and produce Au NCs smaller than Au$_{25}$ NCs (Supplementary Fig. 18). This results from the reduction of Au–MHA complexes free from reactive Au salts. Additionally, we performed the second reduction of highly focused Au$_4$(MHA)$_4$, synthesized by mixing excess amount of MHA with HAuCl$_4$ salts (Supplementary Fig. 19)[48]. Interestingly, no apparent reduction of those complexes to Au NCs occurs even after 7 d of reduction by NaBH$_4$. It can be construed that the excess first reduction, inducing most Au fully surrounded with S, rather hinders further reduction to form Au NCs. Therefore, it is important to optimize the first reduction conditions for the successful synthesis of homogeneous Au NCs.

## Syntheses of metal NCs by controlled complex formation

The effect of the complex formations toward the uniformity of NCs can generally be applied to syntheses of various thiolate-protected metal NCs. The degree of the first reduction also alters the homogeneity of Au$_{25}$ NCs when different Au$^{III}$ salt precursors or thiol ligands are used (Supplementary Fig. 20). When changing Au$^{III}$ salt precursors from HAuCl$_4$ to HAuBr$_4$ (Supplementary Fig. 20a, b), or replacing MHA to MPA (3-mercaptopropionic acid) (Supplementary Fig. 20c, d) or Cys (L-cysteine) (Supplementary Fig. 20e, f), the relationship between the first reduction and the homogeneity of Au NCs is consistently applied. The enhanced homogeneity of Au$_{25}$ NCs is revealed by apparent UV-vis peaks of Au$_{25}$ NCs with high degree of the first reduction (Supplementary Fig. 20a, c, e). Contrarily, a short first reduction time and a small amount of thiol ligands increase the fraction of large Au NCs (Supplementary Fig. 20b, d, f). Although several synthetic conditions of each system are slightly different from those of original Au$_{25}$(MHA)$_{18}$ synthesis, owing to the different reduction potential between HAuCl$_4$ and HAuBr$_4$[49] or different electron density of each thiol ligands (Supplementary Fig. 21), homogeneity of the thiolate-protected Au NCs is controlled by the extent of Au–thiolate complex formation.

More interestingly, we can successfully apply the same mechanistic logic of controlling the metal–thiol interaction to synthesize various types of metal NCs. We could find the optimized conditions for synthesizing homogeneous NCs composed of Ag, Pt, Pd, or Rh (Fig. 5). Two types of atomically precise thiolate-protected Ag NCs, Ag$_{44}$(*p*-MBA)$_{30}$ NCs[20] (*p*-mercaptobenzoic acid) and Ag$_{25}$(MHA)$_{18}$ NCs[50], are uniformly synthesized by high extent of Ag-thiol ligands interaction. In the optimal condition with a large amount of thiol ligands, the two Ag NCs are homogeneously synthesized, as confirmed by UV–vis absorption spectra (Fig. 5a, b), and C$_s$-TEM imaging (Fig. 5f, g). In comparison, using small amount of thiol ligands formation leads to the concurrent synthesis of larger Ag NCs than the two sorts of Ag NCs (Supplementary Fig. 22). In Ag$_{44}$(*p*-MBA)$_{30}$ NC synthesis, the fraction of non-Ag$_{44}$ NCs becomes large with the decreased extent of Ag-(*p*-MBA) interaction (Supplementary Fig. 22b, c). It is also noticeable that low amount of thiol

ligands induces the synthesis of metallic Ag NPs in both cases of $Ag_{44}$ NC and $Ag_{25}$ NC (Supplementary Fig. 22b, d)[51].

It can be further extended to find the optimal synthetic condition for the uniform synthesis of ultrasmall metal NPs composed of non-coinage metals, by controlling the extent of interaction between metal and thiol. We characterized ultrasmall Pt, Pd, and Rh NPs obtained under the optimal conditions of metal–MHA interaction. Successful synthesis of ultrasmall NPs are verified by distinct UV–vis absorption peaks corresponding to quasi-molecular metal NCs (Fig. 5c–e), which do not appear in their metallic NP counterparts[52,53]. $C_s$-TEM imaging also confirms the uniform synthesis of the metal NCs under the optimal conditions (Fig. 5h–j). Such non-coinage metal NCs have similar synthetic trends with the Au or Ag NCs; they obey the relationship between the extent of metal–thiol interaction and homogeneity of metal NCs (Supplementary Figs. 23–25). Using a low amount of MHA ligands and a short reaction time between metal salts and MHA yields a high fraction of relatively large-sized NPs, leading to increase in average size. In contrast, as the amount of MHA or the metal–thiol interaction time increases, the fraction of ultrasmall NPs becomes dominant along with a decrease in average size. It is also noteworthy that, as well as the Au–SR complexes (Supplementary Fig. 9)[25,26] and the Ag–SR complexes (Supplementary Figs. 26, 27)[54], thiol ligands can react with Pt, Pd, and Rh salts to produce metal–thiolate complexes (Supplementary Figs. 28–30)[21,55]. It can be understood that the extent of interaction between MHA ligands and those metal salts is equivalent to the degree of metal–thiolate complex formation. Using small amount of MHA ligands produces poor quality of metal–thiolate complexes compared to those synthesized under the optimal conditions in Fig. 5 (Supplementary Fig. 31). Nonetheless, the existence of strong metal–sulfur bond is essential to synthesize ultrasmall Pt, Pd, or Rh NCs. Unless the thiol ligands coexist with metal salts, the metal precursors are reduced with extremely rapid rate by $NaBH_4$ (Supplementary Fig. 32).

## Discussion

In this study, we investigate the role of metal–thiolate complex formation in regulating the uniform synthesis of metal NCs. In Au NC synthesis, changes in the conditions of complex formation alter the fraction of various Au precursors, directly affecting the homogeneity of Au NCs. The homogeneity of Au NCs is determined by reduction kinetics by $NaBH_4$, which is governed by the coordination chemistry of the Au precursors displaying different reactivity. Such an effect of complex formation on the homogeneity of Au NCs can be generally applied to other synthetic systems of Au NCs, thiolate-protected Ag NCs, and ultrasmall metal NPs composed of non-coinage metal such as Pt, Pd, and Rh. The crucial roles of metal–ligand complex formation studied in this work highlight the importance of precursor chemistry in governing the entire synthetic mechanism and are expected to offer general guidelines for the successful synthesis of various metal NCs.

## Methods

### Materials
Gold(III) chloride trihydrate ($HAuCl_4 \cdot 3H_2O \geq 99.9\%$ trace metals basis), silver nitrate ($AgNO_3$, $\geq 99\%$), sodium tetra-chloropalladate(II) ($Na_2PdCl_4$, 98%), rhodium(III) chloride hydrate ($RhCl_3 \cdot xH_2O$, 99.98% trace metals basis), chloroplatinic acid hexahydrate ($H_2PtCl_6 \cdot 6H_2O$, ACS reagent, $\geq 37.50\%$ Pt basis), gold(I) chloride (AuCl, 99.9% metals basis), gold(III) hydroxide, and sodium aurothiomalate hydrate, 3-mercaptopropionic acid (MPA, $\geq 99\%$), 6-mercaptohexanoic acid (MHA, 90%), L-cysteine (L-Cys, BioUltra, $\geq 98.5\%$ (RT)), sodium borohydride ($NaBH_4$, granular, 99.99 % trace metals basis), methanol (anhydrous, 99.8 %) were purchased from Sigma-Aldrich. Sodium hydroxide (NaOH, bead, >98.0 % (T), EP), hydrochloric acid (HCl, 35.0-37.0 %), ethanol (99.5 %), acetone (99.5 %), and 2-propanol (IPA, 99.5 %)

were purchased from Samchun Chemicals. 4-Mercaptobenzoic acid (p-MBA, >95.0%) was purchased from TCI. All chemicals were used without further purification. Deionized water (18.2 MΩ cm) was used in all procedures.

### Au NC synthesis
To synthesize Au NCs with different Au to SR ratio (1:1, 1:1.5, 1:2, 1:2.5, and 1:3), aqueous solution of $HAuCl_4 \cdot 3H_2O$ (20 mM, 0.5 mL) and aqueous solution of MHA, MPA, or Cys (5 mM, 2, 3, 4, 5, and 6 mL) were mixed in ultrapure $H_2O$ (6.7, 5.7, 4.7, 3.7, and 2.7 mL). Then, an aqueous solution of NaOH (1 M, 0.6 mL) was injected into the reacting solution, followed by stirring. After the desired retention time (from 15 s to 7 days), 0.2 mL solution of $NaBH_4$ (4.3 mg in 1 mL of 0.2 M NaOH) was injected into the solution. After the desired reduction time (from 0 to 3 h), 3 equiv. volume of acetone was added to the solution, and then the mixed solution was centrifuged at 10,000g for 10 min to induce the precipitation of the products. Without additional washing steps, the crude product was redispersed with 10 mL ultrapure $H_2O$ for further characterizations, except for XAS and PAGE experiments. For those two characterizations, the crude product was highly concentrated by dissolving the precipitates with small amounts of $H_2O$. Au–MHA complexes were acquired without the injection of the $NaBH_4$ solution. After the desired retention time of the first reduction, the solution was mixed with 3 equiv. volume of acetone and centrifuged at 10,000g for 10 min to obtain crude precipitation. To separate $[Au(MHA)_2]^-$ complexes from the Au NCs under the different conditions, the as-synthesized Au NCs were purified by ultrafiltration tube with a molecular weight cutoff of 3000 g $mol^{-1}$.

For the cases of reducing $HAu^{III}Cl_4$, $Au^ICl$, $Au^{III}(OH)_3$ and $Au^I$-thiomalate without thiol ligands, aqueous solution of those precursors (20 mM, 0.5 mL) were put in 8.7 mL $H_2O$, followed by the addition of aqueous NaOH solution (1 M, 0.6 mL). After stirring for 15 s, 0.2 mL solution of $NaBH_4$ (4.3 mg in 1 mL of 0.2 M NaOH) solution was injected. For the reduction of focused $Au_4(MHA)_4$ complexes, aqueous solution of $HAuCl_4$ (20 mM, 0.5 ml) and 50 μl of MHA (0.361 mmol) were mixed in 8.7 mL of $H_2O$, followed by the addition of aqueous NaOH solution (1 M, 0.6 mL). After stirring for 15 s, 0.2 mL of the prepared $NaBH_4$ solution was injected.

### Effects of initial reactant concentration and temperature
To synthesize Au NCs with different reactant concentration, aqueous solution of $HAuCl_4$ (10 mM or 40 mM, 0.5 mL) and aqueous solution of MHA (2.5 mM or 10 mM, 4 mL) were mixed in ultrapure $H_2O$ (4.7 mL). Then, an aqueous solution of NaOH (1 M, 0.6 mL) was injected into the reacting solution, followed by stirring. After the 15 s retention time, 0.2 mL solution of $NaBH_4$ (4.3 mg in 1 mL of 0.2 M NaOH) was injected into the solution. The second reduction time was kept for 3 h. To synthesize Au NCs with different temperature, the temperature of ultrapure $H_2O$ (4.7 mL) in vial was initially changed to 0 °C with ice or 50 °C with heating plate. Aqueous solution of $HAuCl_4$ (20 mM, 0.5 mL) and aqueous solution of MHA (5 mM, 4 mL) were then injected into the vial. An aqueous solution of NaOH (1 M, 0.6 mL) was then injected into the reacting solution, followed by stirring. To obtain substantial effect of changing temperature, the duration of first reduction time was kept as 55 min. Then, the temperature was recovered to room temperature in 5 min to prevent changed temperature from affecting the reduction by $NaBH_4$. 0.2 mL solution of $NaBH_4$ (4.3 mg in 1 mL of 0.2 M NaOH) was then injected into the solution. The second reduction time was kept for 3 h. The precipitation process was identical with that described above.

To assess the effect of changed initial reactant concentration, we compared the Au NCs produced from three different reactant concentration (Supplementary Fig. 3). The condition of 15 s of first reduction time and 1:2 Au to SR ratio in Supplementary Fig. 1 was used as standard (original synthesis). $Au_{25}$ NCs were successfully

synthesized regardless of changes in the initial reactant concentration, but the homogeneity of $Au_{25}$ NCs slightly increased with reduced concentration. Meanwhile, the increased reactant concentration neither increased nor decreased the homogeneity of Au NCs. We also assessed the effect of temperature whether it could affect the degree of first reduction (Supplementary Fig. 4). To obtain the substantial effect of changing temperature, we kept the first reduction time as 1 h. The condition of 1 h of first reduction time and 1:2 Au to SR ratio in Supplementary Fig. 1 was used as standard (25 °C). $Au_{25}$ NCs were also successfully produced regardless of changes in temperature. Interestingly, both low and high temperature resulted in slightly higher homogeneity of Au NCs than that produced from the original condition. The moderately elevated temperature is known to enhancing the rate of thiol etching, leading to the formation of homogeneous $Au_{25}$ NCs[35]. The low temperature is known to unifying the size and structure of Au–SR complexes which can help the successful synthesis of homogeneous $Au_{25}$ NCs[34]. Both reactant concentration and temperature can affect the synthetic kinetics and quality of $Au_{25}$ NCs, but these parameters are seemingly less critical toward the extent of first reduction.

## Ag NC synthesis

To synthesize $Ag_{25}$ NCs with different Ag to MHA ratio (2:1, 1:1, and 1:2), aqueous solution of $AgNO_3$ (20 mM, 0.5 mL) and aqueous solution of MHA (5 mM, 1, 2, and 4 mL) were mixed in ultrapure $H_2O$ (8.3, 7.3, and 5.3 mL), followed by stirring. After the desired retention time (for 15 s or 1 day), 0.2 mL solution of $NaBH_4$ (4.3 mg in 1 mL of 0.2 M NaOH) was injected into the solution. After 30 min, the crude solution was characterized by UV·vis absorption spectroscopy and TEM. Ag–MHA complexes were acquired without the injection of the $NaBH_4$ solution. After the desired time of complex formation, the solution was centrifuged at 7,500 g for 1 min to obtain crude precipitation.

To synthesize $Ag_{44}$ NCs with different Ag to p-MBA ratio (1:1, 1:2, and 1:4), aqueous solution of $AgNO_3$ (20 mM, 0.5 mL) and ethanolic solution of p-MBA (50 mM, 0.2, 0.4, and 0.8 mL) were mixed in a mixed solvent of ethanol/water solution (40 vol % of ethanol, total 8.95, 8.75, and 8.35 mL). Then, an aqueous solution of NaOH (1 M, 0.15 mL) was injected into the reacting solution, followed by stirring. After the desired retention time (for 15 s or 1 h), 0.2 mL solution of $NaBH_4$ (4.3 mg in 1 mL of 0.2 M NaOH). After 2 h, the solution was firstly centrifuged at 10,000 g for 1 min to separate $Ag_{44}(p\text{-MBA})_{30}$ from other sorts of Ag NCs. After that, the supernatant containing $Ag_{44}(p\text{-MBA})_{30}$ NCs was characterized by UV–vis absorption spectroscopy and TEM. Ag–(p-MBA) complexes were acquired without the injection of the $NaBH_4$ solution. After the desired time of complex formation, the solution was centrifuged at 7500 g for 1 min to obtain crude precipitation.

## Pt NC synthesis

To synthesize Pt NCs with different Pt to MHA ratio (2:1, 1:1, and 1:1.5), aqueous solution of $H_2PtCl_6 \cdot 6H_2O$ (20 mM, 0.5 mL) and aqueous solution of MHA (5 mM, 1 mL, 2 mL, and 3 mL) were mixed in ultrapure $H_2O$ (7.45 mL, 6.45 mL, and 5.45 mL). Then, an aqueous solution of NaOH (1 M, 0.05 mL) was injected into the reacting solution, followed by stirring. After the desired retention time (for 15 s or 2 h), freshly prepared 1 mL solution of $NaBH_4$ (4.3 mg in 1 mL of $H_2O$) was injected into the solution. After 24 h, 4 equiv. volume of acetone was added to the solution, and then the mixed solution was centrifuged at 10,000 g for 10 min to induce the precipitation of the products. Without additional washing steps, the crude product was redispersed with 10 mL ultrapure $H_2O$ for further characterizations. Pt–MHA complexes were acquired without the injection of the $NaBH_4$ solution. After the desired time of complex formation, the pH of reacting solution was changed to neutral by using 1 M aqueous HCl solution to characterize the Pt–MHA complexes by MALDI-TOF.

## Pd NC synthesis

To synthesize Pd NCs with different Pd to MHA ratio (1:1, 1:1.5, and 1:2), aqueous solution of $Na_2PdCl_4$ (20 mM, 0.5 mL) and aqueous solution of MHA (5 mM, 2 mL, 3 mL, and 4 mL) were mixed in ultrapure $H_2O$ (7.1 mL, 6.1 mL, and 5.1 mL). Then, an aqueous solution of NaOH (1 M, 0.2 mL) was injected into the reacting solution, followed by stirring. After the desired retention time (for 15 s or 7 days), 0.2 mL solution of $NaBH_4$ (4.3 mg in 1 mL of 0.2 M NaOH) was injected into the solution. After 3 h, 3 equiv. volume of acetone was added to the solution, and then the mixed solution was centrifuged at 10,000 g for 10 min to induce the precipitation of the products. Without additional washing steps, the crude product was redispersed with 10 mL ultrapure $H_2O$ for further characterizations. Pd-MHA complexes were acquired without the injection of the $NaBH_4$ solution. After the desired time of complex formation, the solution was mixed with 3 equiv. volume of acetone and then centrifuged at 10,000 g for 10 min to be precipitated.

## Rh NC synthesis

To synthesize Rh NCs with different Rh to MHA ratio (1:1, 1:2, and 1:3), freshly prepared aqueous solution of $RhCl_3 \cdot xH_2O$ (20 mM, 0.5 mL) and aqueous solution of MHA (5 mM, 2 mL, 4 mL, and 6 mL) were mixed in ultrapure $H_2O$ (7 mL, 5 mL, and 3 mL), followed by rapid stirring. After 15 s, freshly prepared 0.5 mL solution of $NaBH_4$ (4.3 mg in 1 mL of $H_2O$) was injected into the solution. After 3 h, the solution was purified by ultrafiltration tube with a molecular weight cutoff of 3000 g $mol^{-1}$. The purified NCs were diluted with 10 mL ultrapure $H_2O$ for further characterizations. Rh–MHA complexes were acquired without the injection of the $NaBH_4$ solution. After the desired time of complex formation, the solution was centrifuged at 7,500 g for 1 min to obtain crude precipitation.

## Characterizations

UV·vis absorption data were acquired via UV–vis spectrophotometer (K LAB Co. Ltd., OPTIZEN POP). The solution was diluted with appropriate solvent to acquire the data, and the optical absorbance was recorded from 200 to 900 nm. For PAGE experiments, concentrated metal NC solutions were resolved by native PAGE performed on a Bio-rad Mini-PROTEAN 3 Cell system. Stacking and separating gels were prepared by diluting acrylamide / bis-acrylamide (29:1) into 5 % and 24.375 %, with adjusting the pH to 6.8 and 8.8 by tris-HCl, respectively. Before loading the samples, empty gel was run in the 0.5X TBE buffer (45 mM tris, 45 mM borate, 1 mM EDTA (pH 8.0)) for 2 h at a constant voltage of 200 V at 4 °C, in order to eliminate remaining ammonium and persulfate ions. After pre-running, samples were mixed with modified 5x sample buffer (25 % glycerol, 60 mM tris-HCl (pH 6.8)) and loaded into the wells, run for 2 h at the same condition with the pre-running. The band of $Au_{25}(MHA)_{18}$ was cut and incubated in $H_2O$ for characterizations of UV·vis absorption and MALDI-TOF. The yield of Au–MHA complexes was obtained by ICP-AES (Shimadzu ICPS-7500). About 2–5 mg samples were dissolved by aqua regia (3 mL of concentrated HCl and $HNO_3$, 3:1 vol: vol, respectively) overnight, and 0.2 g of the acid mixture was diluted with $H_2O$ to have the total mass of 10 g. XAS data of Au L3 edge were acquired from 10 C beamline at Pohang Accelerator Laboratory (PAL), Republic of Korea. The energy of the storage ring electron beam was 2.5 GeV with a ring current of ≈360 mA. The incident X-ray was monochromatized by a Si(111)/Si(311) double-crystal. Samples were concentrated in solution and contained in home-built container. The solution was measured by using transmittance mode. The spectrum is processed by using Athena and Arthemis of Demeter package[56]. The spectrum for a reference Au foil was also measured concurrently to calibrate each sample. For XAS experiment, Au NCs and Au–MHA complexes were highly concentrated by dissolving the washed precipitates using small amount of $H_2O$. Such concentrated solution was held in our home-built holders of which hole is

enough for X-ray to be penetrated. XPS was achieved using Thermo Fisher Scientific ESCALAB 250 at Korea Basic Science Institute (KBSI) Busan center, Republic of Korea. The solution of unwashed Au−MHA complexes was dropped onto the SiO₂ wafer, and the washed Au−MHA complexes were characterized as the status of precipitated solid. DLS experiments were performed by using Malvern Zetasizer Nano. The as-synthesized Au−MHA complexes were diluted with deionized water without washing process.

## Verification of Au NC uniformity via UV−vis absorbance

The fraction of synthesized $Au_{25}$ NCs in Fig. 1 can be indirectly estimated by the sample's optical density at 670 nm ($OD_{670}$). We checked the $OD_{670}$ of homogenous Au NCs synthesized under the conditions of Au: MHA = 1:3, 15 s as 0.057. The solution was five times diluted during UV-vis absorbance. Considering the initial concentration of Au is 1 mM and the molar absorption coefficient of $Au_{25}(SR)_{18}$ NCs at 670 nm is $(8.8 \times 10^3 M^{-1} cm^{-1})$[14,42], the yield of $Au_{25}(SR)_{18}$ NCs under such conditions can be calculated as below:

$$\text{yield} = \{(OD_{670} / \text{molar absorption coefficient}) \cdot (\text{Au atoms in NC}) / (\text{initial [Au]})\}$$
$$= \{(0.057 \times 5 cm^{-1} / 8.8 \times 10^3 M^{-1} cm^{-1}) \cdot 25 / 1 mM\} = 80.97\% \quad (1)$$

The value of 80.97 % is almost identical to the ideal yield of $Au_{25}$ NCs from the stoichiometric equation (79 ± 2 %)[26], which indicates the molecular-level purity of $Au_{25}$ NCs produced from the condition.

## Mass spectrometry

MALDI mass spectra were acquired by using a TOF mass spectrometer (Applied Biosystems Inc., Voyager-DETM STR Biospectrometry Workstation), installed in National Center for Inter-University Research Facilities (NCIRF) at Seoul National University (Seoul, South Korea). For MHA-capped Au, Pd, Rh, Pt complexes and NCs, and MPA-capped Au NCs, concentrated 2,5-dihydroxybenzoic acid (DHB) in solution of H₂O: MeOH = 1:1 was used as a matrix. For Ag−SR complexes, concentrated DHB in solution of H₂O was used as a matrix. For Cys-capped Au NCs, concentrated α-Cyano-4-hydroxycinnamic acid (CHCA) in solution of H₂O: MeOH = 1:1 was used as a matrix. Pulsed nitrogen laser (337 nm, 3 ns pulses) was used for desorption and ionization of Au NCs. 20−25 kV potential accelerated positive ions after desorption and ionization. Electrospray ionization mass spectrometer (ESI-MS) were obtained using ultrahigh-performance liquid chromatography-quadrupole time-of-flight mass spectrometer (SYNAPT G2-Si, Waters) centered at KBSI Seoul center, Republic of Korea. The solution of samples was directly infused into ESI chamber without any treatment. Parameter: sample flow rate 0.05 mL min⁻¹; capillary voltage 2.5 kV; temperature of source and desolvation 120 °C; flow rates of cone and desolvation gas 70 and 200 L h⁻¹, respectively; nebulizer pressure 3 bar.

## TEM

For high-resolution TEM, we firstly synthesized few-layer graphene by the low-pressure chemical vapor deposition (LPCVD) method on a copper foil with a thickness of 30 μm (Nilaco). The copper foil was inserted into a quartz tube and heated to 1,070 °C over 30 min under a 90 atm cm³ min⁻¹ H₂ flow at 2.0 Torr. Under the same gas environment, the copper foil was annealed for 1 h and then cooled to 1,050 °C. The synthesis of graphene was achieved by the subsequent introduction of a gas mixture of 90 atm cm³ min⁻¹ H₂ and 10 atm cm³ min⁻¹ CH₄ at 2.0 Torr for 2 h. After the synthesis completed, fast cooling to room temperature with 90 atm cm³ min⁻¹ H₂ and 10 atm cm³ min⁻¹ CH₄ flow was performed at 2.0 Torr.

Procedures for preparing graphene-coated TEM grids were described as below. One side of a Cu foil, where the both sides were covered with graphene sheets, was milled with sandpaper. Au grids with holey carbon films (Quantifoil) were placed on the graphene-synthesized side of the Cu foil. Placing a drop of isopropyl alcohol and subsequent drying ensured the complete adhesion between the Au grids and the Cu foil. Then, the Cu foil was etched with 1 g 10 mL⁻¹ aqueous ammonium persulfate solution. After the Cu foil was completely etched, the TEM grids floated on the solution were transferred on the DI water and picked up using reverse-action tweezers. The graphene-coated TEM grid was treated with H₂ in reactive ion etcher (RIE, SCIENTIFIC ENGINEERING, Korea), under a 0.8 atm cm³ min⁻¹ H₂ gas flow at 5 W RF power for 15 s. We found that the H₂ plasma treatment removed carbon residues on graphene and enhanced its hydrophilicity. Finally, the 2.5 μL of Au NC solution were dropped onto the grids and then dried at 70 °C for 24 h.

High-resolution TEM was conducted with an JEM-ARM200F (JEOL) operates at 200 kV. The microscope was equipped with a spherical aberration corrector at image-forming lens, a cold field emission gun, and a K3 IS direct electron detector (Gatan), which was installed NCIRF at Seoul National University (Seoul, South Korea). To mitigate electron beam-driven agglomeration between Au NCs, we kept the electron dose rates less than 500 e⁻ Å⁻² s⁻¹ for acquiring each TEM images and the electron beam was blanked except for focusing and capturing the images. The images were treated with ImageJ, and the sizes of the Au NCs were measured from the TEM images.

## Computational details

All density functional theory (DFT) calculations of Au¹−MHA complexes were carried out using ORCA[57] program with a generalized gradient approximation functionals (GGA) suggested by Perdew, Burke, and Ernzerhof (PBE)[58] and Valence triple-zeta polarization basis set (def2-TZVP) basis sets[59]. The Conductor-like Polarizable Continuum Model (C-PCM)[60] was used for considering the solvation effect. The self-consistent field (SCF) convergence threshold was set to $2.72 \times 10^{-7}$ eV. In geometry optimizations, all structures were relaxed to the energy change convergence criterion of $1.36 \times 10^{-4}$ eV, root mean square (RMS) gradient convergence criterion of $5.14 \times 10^{-3}$ eV Å⁻¹, maximum gradient convergence criterion of $1.54 \times 10^{-2}$ eV Å⁻¹, and RMS displacement convergence criterion of $2.12 \times 10^{-3}$ Å. The Au−MHA complexes ranging from one to five Au with MHA ligand, Au⁺ ion, and MHA⁻ ion energies were calculated for estimating $E_{inc}$, $E_{av}$, and $E_{total}$ of $Au_n(MHA)_n$ ($n = 2$–5). For geometry optimization of each thiol ligands, we utilized first-principles density functional theory (DFT) calculations as implemented in General Atomic and Molecular Electronic Structure System (GAMESS)[61,62] with B3LYP/6-31 G** level. The ESP maps of the ligand were calculated from the optimized coordinates and charges of each ligand.

The reason for the higher stability of $Au_{3-5}(MHA)_{3-5}$ than that of $Au_2(MHA)_2$ arises from the cyclic structure, where each Au is coordinated by two S, as confirmed by density functional theory (DFT) calculations. DFT calculations that considered the real MHA ligand showed optimized cyclic structures of $Au_{3-5}(MHA)_{3-5}$ complexes (Fig. 2i−k)[40,41]. However, in the optimized structure of $Au_2(MHA)_2$ complex, there exists one Au coordinated with only one S (Fig. 2h). Based on the energy calculation, the surviving complexes are more stable than Au(MHA) and $Au_2(MHA)_2$, which was confirmed by the incremental fragmentation energy ($E_{inc}$), the average monomer fragmentation energy ($E_{av}$), and the overall dissociation energy ($E_{tot}$), as used in the previous reports (Fig. 2l−n)[41]. The equations for those energies were established as:

$$E_{inc} = E[Au_n(MHA)_n] - E[Au(MHA)] - E[(Au_{n-1}(MHA)_{n-1}] \quad (2)$$

$$E_{av} = \{E[Au_n(MHA)_n] - n \cdot E[(Au(MHA)]\}/n \quad (3)$$

$$E_{\text{total}} = E\left[(Au_n(MHA)_n\right] - n \cdot E[MHA^-] - n \cdot E[Au^+] \qquad (4)$$

The lower energies of $Au_3(MHA)_3$, $Au_4(MHA)_4$, and $Au_5(MHA)_5$ are direct evidence of the enhanced stability of those complexes due to the increased coordination number of Au with S and the resultant closed-ring structure. Such structural differences within Au–MHA complexes directly change their reactivity toward $NaBH_4$ and leadingly the degree of their participation in the synthesis of Au NCs.

## Reporting summary

Further information on research design is available in the Nature Portfolio Reporting Summary linked to this article.

## Data availability

All data are available from the corresponding authors upon request. Source data are provided with this paper.

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

## Acknowledgements

This work was supported by Institute of Basic Science (IBS-R006-D1). J.S.K., S.K., J.K., and J.P. acknowledge a support from the National Research Foundation of Korea (NRF) grant funded by the Korea government (MSIT) (NRF-2020R1A2C2101871 and NRF-2017R1A5A1015365). J.S.K., C.K.S., and J.P. acknowledge a support from Samsung Research Funding & Incubation Center of Samsung Electronics under Project Number SSTF-BA1802-08 for the syntheses and TEM characterizations of metal–thiolate complexes and metal nanoclusters. J.P. acknowledges a support from Samsung Research Funding & Incubation Center of Samsung Electronics under Project Number SRFC-MA2002-03 for simulations and material characterizations. Experiments at the PAL-II 10C Wide Energy EXAFS beamline were supported in part by MEST and POSTECH. We especially thank Kyung Hee Lim at the National Center for Inter-University Research Facilities (NCIRF) for MALDI-TOF MS measurement. We also thank Jinwoo Kim at Korea Basic Science Institute (KBSI) Seoul center and Jong-Seong Bae at KBSI Busan center for ESI-MS and XPS measurement, respectively.

## Author contributions

J.S.K., Hogeun Chang, T.H., and J.P. conceived the original idea, designed the research project and led the project direction. J.S.K. carried out all syntheses, assisted by Hanguk Cho and J.K. J.S.K. participated in all characterization experiments and relevant analyses for metal–thiolate complexes and metal nanoclusters, assisted by Hanguk Cho. S.K. carried out graphene synthesis, and conducted Cs-TEM imaging experiments. S.C. carried out the PAGE experiments of various metal nanoclusters. S.J.K. and N.P. carried out the DFT studies for the optimized structure and energy calculation of Au–SR complexes. Y.K. and C.K.S. assisted and discussed the analyses of XAS data. D.K. carried out the DFT calculation for the ESP maps for different sorts of thiol ligands. J.S.K., Hogeun Chang, T.H., and J.P. wrote the manuscript with the help of the other authors. T.H. and J.P. supervised all aspects of the research.

## Competing interests

The authors declare no competing interests.
