## [Peer Review File · Nature Communications]

Reviewer comments, first round

Reviewer #1 (Remarks to the Author):

In this paper, Kim et al. investigated the role of metal-thiolate complex formation in regulating the uniform synthesis of metal nanoclusters, mainly the Au nanoclusters. For the preparation of gold nanoclusters, the changes in the conditions of the complex formation alter the fraction of various Au precursors, directly affecting the homogeneity of gold nanoclusters. The observed effect of complex formation on the homogeneity of Au nanoclusters can be generally applied to other systems, including other Au nanoclusters or Ag/Pt/Pd/Rh nanoclusters. The reported evaluations for the preparation of homogeneous metal nanoclusters are interesting, and should have some implications for the synthetic chemistry of metal nanoclusters. However, some necessary discussions and experiments are lacking in its current form. Thus, I would like to suggest a major revision of this paper.

(1) For the effect of gold complex components on the homogeneity and the synthetic yield of the Au₂₅ cluster, more careful controls are needed. For the discussion "a longer first reduction time or a more amount of MHA produces more homogeneous Au₂₅ NCs without the formation of additional NCs", there should be more underlying chemistry for the formation of Au₂₅. Besides, in the "Effect of the first reduction on the fraction of various Au-thiolate complexes" part, they further demonstrated the importance of some Au-MHA complexes for the generation of homogeneous Au₂₅ nanoclusters. Is that possible to further control the Au-MHA complexes before the second reduction, i.e., using PAGE or HPLC to extract some Au-MHA complexes that are important for the Au₂₅ formation, and accomplishing the synthesis of Au₂₅ clusters via stoichiometry, with which way the necessity of these Au-MHA complexes for the formation of Au₂₅ can be further demonstrated.

(2) The dynamic light scattering characterization is necessary here for elevating the homogeneity of as-synthesized nanoclusters and the metal complex precursors. The ESI and TEM results can not represent the overall compositions in solutions, while a combination of ESI, TEM, and DLS could be more feasible.

(3) The Au-based intermediate cluster products should be considered in evaluating the effect of the coordination of Au species on the kinetics of the second reduction. In the second reduction process, the Au₂₅ nanocluster with eight free electrons (8e) was not generated directly, and some intermediate cluster products with 2e, 4e, 6e, and even 10e, should exist, and would be affected by the reduction rate. The dynamic factor of these intermediate cluster products should be considered for the formation of homogeneous Au₂₅ nanoclusters.

Reviewer #2 (Remarks to the Author):

The synthesis involving metal thiolate complexes is the key protocol to produce most of thiolate-protected metal nanoclusters, including the representative Au and Ag nanoclusters. Although the M-thiolate complexes produced from reaction between metal salts and thiols are direct precursor identified in the reduction reaction to final products, the whole synthetic systems are somehow too 'dirty' to find out the true relationship between distribution of precursor with the homogeneity of metal nanoclusters. In this paper, the authors successfully identified this key issue existing in the synthesis of thiolate-protected metal nanoclusters and used Au₂₅ nanocluster as the model system to find out the relationship between precursor and final product. Generally, I think this research is important, especially for discovery more methods to produce different kinds of metal nanocluster species. I am also exciting to see that the similar principle can be applied to Pt, Pd and Rh clusters. I support its publication after addressing the following a few concerns from me.

1. For small Au-thiolate complexes and final products, maybe ESI-MS is a 'softer' method to get more precise information on their distribution and more sensitive than MOLDI-MS. Have you tried it and get some data?

2. How did you prepare the samples for EXAFS. Did you dry the sample? The handling of samples

may change the precursor as they are so sensitive to the chemical environment.

3. What are the approximate sizes of Pt, Pd and Rh clusters produced using MHA as ligand?

4. Have you observed the AuMHA₂ complex in your system? It is a major species involved in the growth process of Au nanoclusters.

Reviewer #3 (Remarks to the Author):

The authors have examined the detailed processes of metal nanocluster synthesis and concluded that the metal-ligand complexes play critical roles. However, this conclusion is well-known in the community, especially, from the work of Jianping Xie, et al. (e.g., their recent work <https://doi.org/10.1016/j.xcrp.2020.100206>). Therefore, I don't think the present work is particularly novel and exciting. In addition, researchers have now been able to routinely obtain molecularly pure gold nanoclusters and their single crystals, so the impact of this work is limited in terms of control the homogeneity of final products.

Reviewer #4 (Remarks to the Author):

This manuscript reported a very comprehensive study on the influence of the precursor metal-ligand complexes in the metal (Ag, Au) NC synthesis. By controlling the fraction of Au species showing different reactivity, the authors found that the extent of complex formation alters reduction kinetics and the homogeneity of Au NCs, and this trend can be universally applied to syntheses of Ag, Pt, Pd, and Rh NCs. I recommend the publication of this work on Nat. Comm. after considering follow revisions:

1. The major point of this work, the homogeneity of metal-ligand complexes before the reduction affects homogeneity of product metal cluster, has been previous proposed by Zhu and Jin in their report of synthesis of Au₂₅ cluster [JACS, 2008]. The author should aware this.
2. Discussions should be added about how the metal-ligand complexes were converted into the homogeneous metal NCs upon the NaBH₄ reduction. For instance, some mechanisms were recently reported for the formation of Au_n(SR)_m clusters from the specific Au-SR oligomers [The Journal of Physical Chemistry C 2022, 126, 5980-5990; Journal of the American Chemical Society 2021, 143, 15224-15232; Small 2021, 17, 2000627].
3. In the final part of the work, the strategy was used to synthesize the Pt, Pd and Rh clusters. Did the exact molecular formula of the Pt-SR, Pd-SR and Rh-SR clusters determined by the MS? In addition, from the absorption curve, no molecular like feature absorption peak was observed for the Pt-SR, Pd-SR and Rh-SR clusters (Fig. 5)?

Responses to the reviewer's comments

The followings are the responses to the reviewers' comments for the manuscript (NCOMMS-22-47215), entitled "Critical roles of metal-ligand complexes for synthesizing metal nanoclusters".

Reviewer: 1

General Comments: In this paper, Kim et al. investigated the role of metal-thiolate complex formation in regulating the uniform synthesis of metal nanoclusters, mainly the Au nanoclusters. For the preparation of gold nanoclusters, the changes in the conditions of the complex formation alter the fraction of various Au precursors, directly affecting the homogeneity of gold nanoclusters. The observed effect of complex formation on the homogeneity of Au nanoclusters can be generally applied to other systems, including other Au nanoclusters or Ag/Pt/Pd/Rh nanoclusters. The reported evaluations for the preparation of homogeneous metal nanoclusters are interesting, and should have some implications for the synthetic chemistry of metal nanoclusters. However, some necessary discussions and experiments are lacking in its current form. Thus, I would like to suggest a major revision of this paper.

Response: We appreciate the reviewer for the careful evaluations. In agreement with his or her comments, we have revised the manuscript in point-by-point manner.

Comment 1: For the effect of gold complex components on the homogeneity and the synthetic yield of the Au₂₅ cluster, more careful controls are needed. For the discussion "a longer first reduction time or a more amount of MHA produces more homogeneous Au₂₅ NCs without the formation of additional NCs", there should be more underlying chemistry for the formation of Au₂₅. Besides, in the "Effect of the first reduction on the fraction of various Au-thiolate complexes" part, they further demonstrated the importance of some Au-MHA complexes for the generation of homogeneous Au₂₅ nanoclusters. Is that possible to further control the Au-MHA complexes before the second reduction, i.e., using PAGE or HPLC to extract some Au-MHA complexes that are important for the Au₂₅ formation, and accomplishing the synthesis of Au₂₅ clusters via stoichiometry, with which way the necessity of these Au-MHA complexes for the formation of Au₂₅ can be further demonstrated.

Response 1-1: We appreciate the reviewer for the valuable questions. To assess the effect of gold complexes on the homogeneity of Au NCs more carefully, we performed additional control experiments under different synthetic conditions. Other than the first reduction time or the ratio of Au to MHA, we tried changing reactant concentration and reaction temperature which can affect the extent of complex formation and monodispersity of Au NCs. Firstly, we compared the Au NCs produced from three different reactant concentration under the same condition of 15 seconds of first reduction time and 1:2 Au to MHA ratio (**Fig. R1.1**). Au₂₅ NCs

were successfully synthesized regardless of changes in the initial reactant concentration, but the purity of Au₂₅ NCs slightly increased with the reduced initial concentration, while the increased reactant concentration did not considerably hamper the homogeneity. We also assessed the effect of temperature on the homogeneity of Au NCs under the same condition of 1 hour of first reduction time and 1:2 Au to MHA ratio (**Fig. R1.2**). Both low and high temperature resulted in more uniform Au₂₅ NCs than those produced under 25°C. The moderately elevated temperature is known to enhancing the rate of thiol etching, leading to the formation of homogeneous Au₂₅ NCs [*J. Phys. Chem. C* **121**, 20, 10743–10751 (2017)]. The low temperature is known to unifying the size and structure of Au-SR complexes which contributes to successful synthesis of homogeneous Au₂₅ NCs [*J. Am. Chem. Soc.* **130**, 1138–1139 (2008)].

Both reactant concentration and temperature can affect the synthesis and quality of Au₂₅ NCs, but these parameters are seemingly less effective for controlling the Au-SR complexes. In other words, it can support our results that first reduction time and ratio of MHA to Au are the main parameters affecting the extent of complex formation. We added one paragraph in revised manuscript and one supplementary note in supplementary information describing the effect of reactant concentration and temperature. We added the two references regarding the effect of temperature on the Au NC synthesis. We also added **Fig. R1.1** and **Fig. R1.2** described above as **Supplementary Fig. 3,4**, respectively in the revised supplementary information. The synthetic procedures with changed concentration and temperature were also added in **Syntheses** part in revised manuscript.

Fig. R1.1 (Supplementary Fig. 3 in revised SI). **a**, UV-Vis spectra, and **b**, MALDI-TOF mass spectra (in the linear mode) of Au NCs synthesized under three conditions of different initial reactant concentration. The ratio of Au precursor to MHA is fixed as 1:2, the first reduction time is fixed as 15 seconds, and second reduction time under all conditions is fixed as 3 hours. Dashed lines indicate the existence of Au₂₅ NCs in both **a** and **b**. Arrows in **b** indicate the mass of Au NCs bigger than Au₂₅(MHA)₁₈. The data of original synthesis is matched with those in **Fig 1**.

Fig. R1.2 (Supplementary Fig. 4 in revised SI). **a**, UV-Vis spectra, and **b**, MALDI-TOF mass spectra (in the linear mode) of Au NCs synthesized under three conditions of different temperature. The ratio of Au precursor to MHA is fixed as 1:2, the first reduction time is fixed as 1 hour, and second reduction time under all conditions is fixed as 3 hours. Dashed lines indicate the existence of Au₂₅ NCs in both **a** and **b**. The data of 25°C is matched with those in **Supplementary Fig. 1**.

(Page 6, line 96; in the revised manuscript)

Other than the two parameters, there exist other parameters, reactant concentration or temperature, which can control the Au NC synthesis^{34,35}. We examined how they affect the first reduction or homogeneity of produced Au NCs (**Supplementary Fig. 3,4**). Concentration or temperature shows less effect on both the degree of first reduction and Au NC homogeneity than the first reduction time and the ratio of Au precursor to thiol ligands (**Supplementary note 1**).

(Syntheses; in the revised manuscript)

To synthesize Au NCs with different reactant concentration, aqueous solution of HAuCl₄ (10 mM or 40 mM, 0.5 ml) and aqueous solution of MHA (2.5 mM or 10 mM, 4 ml) were mixed in ultrapure H₂O (4.7 ml). Then, an aqueous solution of NaOH (1 M, 0.6 ml) was injected into the reacting solution, followed by stirring. After the 15 seconds retention time, 0.2 ml solution of NaBH₄ (4.3 mg in 1 ml of 0.2 M NaOH) was injected into the solution. The second reduction time was kept for 3 hours. To synthesize Au NCs with different temperature, the temperature of ultrapure H₂O (4.7 ml) in vial was initially changed to 0 °C with ice or 50 °C with heating plate. Aqueous solution of HAuCl₄ (20 mM, 0.5 ml) and aqueous solution of MHA (5 mM, 4 ml) were then injected into the vial. An aqueous solution of NaOH (1 M, 0.6 ml) was then injected into the reacting solution, followed by stirring. To obtain substantial effect of changing

temperature, the duration of first reduction time was kept as 55 minutes. Then, the temperature was recovered to room temperature in 5 minutes to prevent changed temperature from affecting the reduction by NaBH₄. 0.2 ml solution of NaBH₄ (4.3 mg in 1 ml of 0.2 M NaOH) was then injected into the solution. The second reduction time was kept for 3 hours. The precipitation process was identical with that described above.

(Newly added references; in the manuscript)

34. Zhu, M., Lanni, E., Garg, N., Bier, M. E. & Jin, R. Kinetically controlled, high-yield synthesis of Au₂₅ clusters. *J. Am. Chem. Soc.* **130**, 1138–1139 (2008).
35. Chen, T., Yao, Q., Yuan, X., Nasaruddin, R. R. & Xie, J. Heating or Cooling: Temperature Effects on the Synthesis of Atomically Precise Gold Nanoclusters. *J. Phys. Chem. C* **121**, 10743–10751 (2017).

(Supplementary note 1; in the supplementary information)

Discussion on effects caused by changes in initial reactant concentration and temperature

To assess the effect of changed initial reactant concentration, we compared the Au NCs produced from three different reactant concentration (**Supplementary Fig. 3**). The condition of 15 seconds of first reduction time and 1:2 Au to SR ratio in Figure S1 was used as standard (original synthesis). Au₂₅ NCs were successfully synthesized regardless of changes in the initial reactant concentration, but the homogeneity of Au₂₅ NCs slightly increased with reduced concentration. Meanwhile, the increased reactant concentration neither increased nor decreased the homogeneity of Au NCs. We also assessed the effect of temperature whether it could affect the degree of first reduction (**Supplementary Fig. 4**). To obtain the substantial effect of changing temperature, we kept the first reduction time as 1 hour. The condition of 1 hour of first reduction time and 1:2 Au to SR ratio in **Supplementary Fig. 1** was used as standard (25 °C). Au₂₅ NCs were also successfully produced regardless of changes in temperature. Interestingly, both low and high temperature resulted in slightly higher homogeneity of Au NCs than that produced from the original condition. The moderately elevated temperature is known to enhancing the rate of thiol etching, leading to the formation of homogeneous Au₂₅ NCs⁹. The low temperature is known to unifying the size and structure of Au-SR complexes which can help the successful synthesis of homogeneous Au₂₅ NCs¹⁰. Both reactant concentration and temperature can affect the synthetic kinetics and quality of Au₂₅ NCs, but these parameters are seemingly less critical toward the extent of first reduction.

Response 1-2: For the discussion “a longer first reduction time or a more amount of MHA produces more homogeneous Au₂₅ NCs without the formation of additional NCs” in page 6, the effects of those two main parameters on the extent of Au-SR complex formation and homogeneity of Au NCs are sequentially explained in **Fig. 2–4**. The long first reduction time or the high ratio of MHA to Au lead to synthesize large amount of Au-MHA complex with high coordination number of Au-S bond, as described in **Fig. 2b~g**. Those complexes possess high

stability in terms of coordination chemistry as confirmed by DFT calculation in **Fig. 2h~n**. Their stability renders the kinetics of 2nd reduction by NaBH₄ slow, which is prerequisite for the controlled synthesis of homogeneous Au NCs [*Coord. Chem. Rev.* **329**, 1–15 (2016)], as described in **Fig. 4**. In contrast, short first reduction time and small amount of MHA yields small fraction of Au-MHA complex with low coordination number of Au-S bond, resulting in rapid reduction by NaBH₄ and polydisperse Au NCs as described in **Fig. 3**. For helping readers further understand the logic in this research, we added schemes summarizing the different synthetic mechanism from Au-SR complexes to Au NCs in **Fig. 3,4** respectively. We also added descriptions for these schemes in the revised manuscript.

(**Fig. 3**; in the revised manuscript)

Fig. 3. Fast second reduction kinetics and less homogeneous Au NCs. a,d, UV-Vis absorption spectra, b,e, MALDI-TOF mass spectra in the linear mode, and c,f, Digital photographs of Au NCs during the second reduction using precursors obtained applying the 1:1, 15 s, and the 1:2, 15 s conditions, respectively. Dashed lines in UV-Vis absorption and

mass spectra indicate the evidence of Au₂₅ NCs. Arrows in **a** indicate the decreasing fraction of Au₂₅ NCs with increasing second reduction time, while those in **d** indicate the increasing fraction of Au₂₅ NCs with increasing second reduction time. Arrows in **b,e** indicate the existence of Au NCs larger than Au₂₅. **g**, Reaction mechanism scheme when the degree of the first reduction is low.

(Fig. 4; in the revised manuscript)

Fig. 4. Slow second reduction kinetics and more homogeneous Au NCs. **a,d**, UV-Vis absorption spectra, **b,e**, MALDI-TOF mass spectra in the linear mode, and **c,f**, Digital photographs of Au NCs during the second reduction using precursors obtained applying 1:3, 15 s and 1:2, 7 d conditions, respectively. Dashed lines in UV-Vis absorption and mass spectra indicate the evidence of Au₂₅ NCs. Arrows in **a,d** indicate an increasing fraction of Au₂₅ with increasing second reduction time. Arrows in **b,e** indicate the existence of Au NCs smaller than Au₂₅. Evidence of Au NCs smaller than Au₂₅ also appears in UV-Vis absorption spectra (black asterisks). **g**, Reaction mechanism schematic when the degree of the first reduction is high.

(Page 11, line 212; in the revised manuscript)

In summary, a schematic of the synthetic mechanism of less homogeneous Au NCs is proposed in Fig. 3g. Low degree of first reduction yields a small fraction of Au-MHA complexes, and the remaining Au salts majorly participate in the formation of Au NCs. The high reactivity of Au salts and the Au-MHA complexes, with relatively low Au-S coordination, induces fast second reduction kinetics and results in the synthesis of Au NCs with reduced homogeneity.

(Page 11, line 227; in the revised manuscript)

To summarize the synthetic mechanism as displayed in Fig. 4g, high degree of the first reduction consumes most of the Au salts, producing Au-MHA complexes with high Au-S coordination as the major species. The stability of these complexes enables the controlled synthesis of homogeneous Au₂₅ NCs driven by slow second reduction kinetics.

Response 1-3: As the reviewer requests, we tried to separate Au-MHA complexes by using PAGE experiment and react those with NaBH₄ in terms of stoichiometry. However, we figure out that separating Au-MHA complexes via PAGE is not trivial. We could not detect any sign of separated complexes in each lane in PAGE as shown in Fig. R1.3, even though we highly concentrated the aqueous Au-MHA complexes solution. There are two reasons for the difficulty separating Au-MHA complexes via PAGE experiments; **i)** Au-MHA complexes in solution display too pale color to be detected in PAGE lane, and **ii)** the mass of the complexes are too similar to be separated as the band merging is severe, as reported in a previous report [*Nat. Commun.* **12**, 3212 (2021)]. Instead, combining the yield of the complexes from HAuCl₄ under the synthetic conditions in Fig. 2b and the intensity of each Au-MHA complexes detected in mass spectra in Fig. 2a, we can indirectly calculate the approximate amount of each complexes as Table R1.1. Considering the Au₄(MHA)₄ is the main complex under all conditions, we may guess a stoichiometric equation with Au₄(MHA)₄ and the electrons generated by NaBH₄ as reactants, and Au₂₅(MHA)₁₈ NC and Au(MHA)₂ complex as product like the equation R1 [*J. Am. Chem. Soc.* **140**, 11370–11377 (2018)].

Under the conditions of high extent of complex formation in Fig. 4, the amount of electrons is the eight times of that of NaBH₄ in this study: 8 * 112 mM * 0.2 ml = 179.2 μmol [*Adv. Mater.* **32**, 9, 1906063 (2020)]. Under the conditions of low extent of complex formation, we can synthesize more homogeneous Au₂₅(MHA)₁₈ NCs than those in Fig. 3 by using relatively small amount of NaBH₄ as shown in Fig. R1.4,5. The appropriate concentration of NaBH₄ under the condition of Au : MHA = 1:1 and 15 seconds of first reduction time lies between 5 mM and 6 mM (Fig. R1.4), while that under the condition of Au : MHA = 1:2 and 15 seconds of first reduction time is presumably 100 mM (Fig. R1.5). However, we tried to solely focus on controlling the metal-thiolate complexes and assessing their effects on the metal

NC synthesis throughout this study. Thus, we did not include the data from the experiment using different amount of NaBH₄.

Meanwhile, it is noteworthy that the amount of NaBH₄ in this whole study far exceeds the previously known ‘stoichiometric amount of NaBH₄’ for synthesizing Au₂₅(*p*-MBA)₁₈ NCs [*J. Am. Chem. Soc.* **140**, 11370–11377 (2018)]. Even the reduced amount of NaBH₄ under the conditions of low extent of Au-MHA complex formation (**Fig. R1.4,5**) is more than the stoichiometric amount. Therefore, it is hard to establish an accurate stoichiometric equation by only considering the amount of NaBH₄ and Au-MHA complexes. Such trend can arise from the different stability of Au-SR complexes. It is known that the Au-SR complexes with aromatic thiol ligands have lower stability than those with aliphatic thiol ligands [*J. Phys. Chem. A* **115**, 11818–11823 (2011)], which may change the amount of NaBH₄ to reduce the complexes to NCs. The differences in Au NC synthesis caused by utilizing different sorts of thiol ligands will be an interesting topic to research, but we think that it is beyond the scope of the current research.

Fig. R1.3 (For reviewer only). (Upper) Digital photograph of highly concentrated aqueous Au-MHA complexes in **Fig. 2**. (Lower) Digital photograph of PAGE gel separating the aqueous Au-MHA complexes showing no apparent bands.

Table R1.1 (For reviewer only). Amount of each Au-MHA complexes detected as **Supplementary Fig. 9** in revised SI, produced under the conditions of **Fig. 2**.

	Au ₂ (MHA) ₂	Au ₃ (MHA) ₂	Au ₃ (MHA) ₃	Au ₄ (MHA) ₃	Au ₄ (MHA) ₄	Au ₅ (MHA) ₅
Au : MHA = 1:1 15 s	0.011 μmol (5.39 %)	0.0067 μmol (4.96 %)	0.04 μmol (2.95 %)	0.085 μmol (8.46 %)	0.74 μmol (74.7 %)	0.028 μmol (3.58 %)

Au : MHA = 1:2 15 s	0.022 μmol (6.43 %)	0.0073 μmol (3.14 %)	0.0083 μmol (3.71 %)	0.0048 μmol (2.83 %)	1.38 μmol (80.3 %)	0.0048 μmol (3.57 %)
Au : MHA = 1:3 15 s	0.017 μmol (4.2 %)	0.0033 μmol (1.34 %)	0.0077 μmol (2.89 %)	0.0028 μmol (1.35 %)	1.71 μmol (86.8 %)	0.0054 μmol (3.41 %)
Au : MHA = 1:2 7 d	0.02 μmol (5.24 %)	0.006 μmol (2.34 %)	0.0093 μmol (3.56 %)	0.0072 μmol (3.72 %)	1.56 μmol (81.0 %)	0.0064 μmol (4.14 %)

Fig. R1.4 (For reviewer only). **a**, UV-Vis spectra, and **b**, MALDI-TOF mass spectra (in the linear mode) of Au NCs synthesized under three conditions of different amount of NaBH₄. The ratio of Au precursor to MHA is fixed as 1:1, the first reduction time is fixed as 15 seconds, and second reduction time under all conditions is fixed as 3 hours. Dashed lines indicate the existence of Au₂₅ NCs in **b**. Arrows in **b** indicate the mass of Au NCs bigger than Au₂₅(MHA)₁₈. The data of original synthesis is matched with those in **Fig. 1**.

Fig. R1.5 (For reviewer only). **a**, UV-Vis spectra, and **b**, MALDI-TOF mass spectra (in the linear mode) of Au NCs synthesized under two conditions of different amount of NaBH₄. The ratio of Au precursor to MHA is fixed as 1:2, the first reduction time is fixed as 15 seconds, and second reduction time under all conditions is fixed as 3 hours. Dashed lines indicate the

existence of Au₂₅ NCs in both **a** and **b**. Arrows in **b** indicate the mass of Au NCs bigger than Au₂₅(MHA)₁₈. The data of original synthesis is matched with those in **Fig. 1**.

Comment 2: The dynamic light scattering characterization is necessary here for elevating the homogeneity of as-synthesized nanoclusters and the metal complex precursors. The ESI and TEM results can not represent the overall compositions in solutions, while a combination of ESI, TEM, and DLS could be more feasible.

Response 2: We appreciate the reviewer's helpful comments. We performed the DLS experiment for the as-synthesized Au nanoclusters without washing procedure as shown in **Fig. R1.6**. The size distribution of Au NCs under the different conditions matched well with the information from other characterizations including mass spectra and TEM, except for that of Au NCs under the condition of Au : MHA = 1:2 and 7 days of first reduction time. Au NCs under that condition showed bigger size in DLS data than that from other characterization methods. However, considering that no plasmonic absorbance peak appears from such Au NC (**Fig. 1a**) it does not indicate that such-sized metallic Au NPs are synthesized under the condition. Instead, small Au NCs under the synthetic condition seemingly form aggregates of the 3~4 nm size, of which possibility is supported by previous reports [*Science* **345**, 6199, 909-912 (2014)], [*Angew. Chem. Int. Ed.* e202215746 (2023)]. We tried to obtain the hydrodynamic diameter of such Au NCs in single-cluster level by diluting them by 1 M NaOH aqueous solution instead of H₂O, but it did not help acquire the desired size by DLS.

Moreover, we acquired the DLS data of Au-MHA complexes under the conditions of **Fig. 2** as shown in **Fig. R1.7**. Interestingly, the data of aggregated Au-MHA complexes under different conditions show a single distribution with similar size, which is known to be a requirement for synthesizing homogeneous Au NCs [*J. Am. Chem. Soc.* **130**, 1138–1139 (2008)]. This can support the importance of our study that the Au-SR complexes should be deeply investigated by several characterizations to assess their effect on the homogeneity of Au NCs. We added the DLS data of **Fig. R1.7** as **Supplementary Fig. 10** in the revised supplementary information. We also added descriptions for the DLS data of Au-SR complexes in the revised manuscript. The information of DLS equipment was also included in the **Supplementary Methods** part in the revised supplementary information.

Fig. R1.6 (For reviewer only). DLS spectrum of as-synthesized Au NCs under each conditions. The colors of samples are matched with the data in **Fig. 1**.

Fig. R1.7 (Supplementary Fig. 10 in revised SI). DLS spectrum of aggregated Au-MHA complexes under each conditions. The colors of samples are matched with the data in **Fig. 2**.

(Page 8, line 144; in the revised manuscript)

The compositions of the detected complexes under each condition are seemingly similar. Additionally, those complexes in solution state form single-sized aggregation with similar hydrodynamic diameter investigated by dynamic light scattering (DLS), regardless of synthetic conditions (Supplementary Fig. 10)³⁴.

(Supplementary Methods; in the supplementary information)

Dynamic light scattering (DLS)

DLS experiments were performed by using Malvern Zetasizer Nano. The as-synthesized Au-MHA complexes were diluted with deionized water without washing process.

Comment 3: The Au-based intermediate cluster products should be considered in evaluating the effect of the coordination of Au species on the kinetics of the second reduction. In the second reduction process, the Au₂₅ nanocluster with eight free electrons (8e) was not generated directly, and some intermediate cluster products with 2e, 4e, 6e, and even 10e, should exist, and would be affected by the reduction rate. The dynamic factor of these intermediate cluster products should be considered for the formation of homogeneous Au₂₅ nanoclusters.

Response 3: We appreciate the important comments from the reviewer. To begin with the synthesis of Au NCs from high extent of complex formation in **Fig. 4**, there exist the intermediate clusters with various free electrons, which were described as ‘smaller-sized NCs’ in the original manuscript. Such smaller-sized NCs can be matched as specific intermediate Au NCs which were also detected in previous studies [*J. Am. Chem. Soc.* **136**, 10577–10580 (2014)], [*J. Am. Chem. Soc.* **140**, 11370–11377 (2018)], as shown in **Fig. R1.8**. Indeed, considering the prominent peak at mass spectra from Au NCs with the second reduction time of 1 hour, the UV-Vis peak at 520 nm of the Au NCs in **Fig. 4a,d** presumably originates from Au₁₈(MHA)₁₄. It is known that there exist specific Au₁₈(SR)₁₄ isomers which absorbs 520 nm light, proved by both experiments and DFT calculations in several studies [*J. Phys. Chem. Lett.* **3**, 1997–2002 (2012)], [*Phys. Chem. Chem. Phys.* **14**, 3737–3740 (2012)].

Moreover, for the synthesis under the condition of Au : MHA = 1:2 and 15 seconds of first reduction time in **Fig. 3d~f**, several intermediate clusters occur throughout the whole second reduction time as displayed in **Fig. R1.9**. However, Au₂₅(MHA)₁₈ NC is also concurrently formed with such intermediates right after injecting the NaBH₄. Considering the mass of bigger clusters than Au₂₅(MHA)₁₈ NCs in the final product under the condition in **Fig. 3d~f** is matched with that of Au₃₈(MHA)₂₄ NCs, such trend resembles the synthesis of Au₂₅(p-MBA)₁₈ NCs by using more ‘over-stoichiometric amount’ of NaBH₄ in the previous study [*J. Am. Chem. Soc.* **140**, 11370–11377 (2018)]. Thus, under the condition of Au : MHA = 1:2 and 15 seconds of first reduction time, it is difficult to separately detect the formation of Au₂₅(MHA)₁₈ NCs from that of intermediate NCs due to the fast reduction kinetics.

Such kinetic becomes too rapid to observe even the intermediate clusters during the reduction of Au-MHA complexes to Au NCs under the condition of **Fig. 3a~c**. Even when reducing the amount of NaBH₄ to the ‘stoichiometric amount’ under the same condition of **Fig. 3a~c**, Au₂₅(MHA)₁₈ NCs are concurrently synthesized with various sorts of intermediate NCs like **Fig. R1.9** (**Fig. R1.10**). Considering our argument that remaining Au salts act as major species participating in the Au NC synthesis when the extent of Au-SR complex formation is low, we can infer that Au₂₅ NCs under such condition are synthesized via different route from the continuous growth of Au-SR complexes or intermediate NCs. We obtained the solution of byproduct from that of as-synthesized Au NCs in **Fig. 1** by using ultrafiltration tube and performed electrospray ionization mass spectrometry (ESI-MS) experiments (**Fig. R1.11**).

Interestingly, only the solution from the condition of Au : MHA = 1:1 and 15 seconds of first reduction time does not contain $[\text{Au}(\text{MHA})_2]^-$ complex, which is well-known byproduct of Au₂₅ NC synthesis from the reduction of Au-SR complexes [*J. Am. Chem. Soc.* **140**, 11370–11377 (2018)]. Based on these additional experiments, we can add two important statements in this research; **i)** the extent of Au-SR complex formation can even govern the emergence of intermediate Au NCs during the second reduction process, and **ii)** the insufficient amount of Au-SR complex makes the synthetic route of Au NCs different from the previously known reaction pathway. We added sentences and revised the previous descriptions regarding the intermediate Au NCs and the emergence of $[\text{Au}(\text{MHA})_2]^-$ complexes in the revised manuscript. We added the references which explains the UV-Vis absorbance of intermediate Au NCs. We also added **Fig. R1.8**, **Fig. R1.9**, and **Fig. R1.11** as **Supplementary Fig. 15,14, and 13**, respectively, in the revised supplementary information. Moreover, procedure for obtaining $[\text{Au}(\text{MHA})_2]^-$ complex was included in the **Syntheses** part in revised manuscript, and that for electrospray ionization mass spectra was added in the **Supplementary Methods** part in the revised supplementary information.

Fig. R1.8 (Supplementary Fig. 15 in revised SI). **a,b**, Zoom-in mass spectra of intermediate Au NCs in Fig. 4a~c and Fig. 4d~f, respectively. Dashed lines indicate the intermediate Au NCs labeled with numbers. **c**, Table of the molecular formulas labeled NCs in **a,b** with the number of free valence electrons in parentheses.

Fig. R1.9 (Supplementary Fig. 14 in revised SI). (Left) Zoom-in mass spectra of intermediate Au NCs in Fig. 3d~f. Dashed lines indicate the intermediate Au NCs labeled with numbers. (Right) Table of the molecular formulas labeled NCs in mass spectra with the number of free valence electrons in parentheses.

Fig. R1.10 (For reviewer only). (Left) Zoom-in mass spectra of Au NCs synthesized under the condition of Au : MHA = 1:1, 15 seconds of first reduction time, and 1.5 mM of NaBH_4 solution. Dashed lines indicate the intermediate Au NCs labeled with numbers. (Right) Table of the molecular formulas labeled NCs in mass spectra with the number of free valence electrons in parentheses.

Fig. R1.11 (Supplementary Fig. 13 in revised SI). a, Digital photograph of separated solution of side product (bottom) and concentrated Au NC solutions in Fig. 1, Experimental and calculated (black) ESI mass spectra of $[\text{Au}(\text{MHA})_2]^-$ complex in the filtrated solution under the different conditions. All colors are corresponding to those in Fig. 1.

(Page 10, line 181; in the revised manuscript)

In contrast, the intensity of Au-S peak at 1.87 Å considerably decreases in the other two cases (Fig. 2f,g), implying that the use of Au-MHA complexes enhances the homogeneity of Au₂₅ NCs in the second reduction. In addition, the solution of Au NCs synthesized under the condition of Au : MHA = 1:1 and 15 seconds of first reduction time does not contain $[\text{Au}(\text{MHA})_2]^-$ complex, the well-known byproduct from the reduction reaction of Au-SR complexes to Au₂₅ NCs (Supplementary Fig. 13)²⁶. It infers that, when the extent of complex formation is low, Au NCs are synthesized from different synthetic route where the starting precursors are different from Au-SR complexes.

(Page 10, line 202; in the revised manuscript)

(before)

This trend is due to the focusing process of polydispersed Au NCs by the enhanced etching ability of thiol ligands under basic condition³³, which is also revealed in the mass spectra of Au NCs (Fig. 3e). Nonetheless, the homogeneity of the final product in this condition is not optimal because of the existence of NCs larger than Au₂₅ NCs. In addition, the formation of Au NCs is rapidly initiated immediately following the NaBH₄ injection, as indicated by abrupt color change under both conditions (Fig. 3c.f).

(after)

The intermediate Au NCs are concurrently formed with Au₂₅ NCs after the initiation of second reduction²⁶, but their fraction decreases along with the reduction time (**Fig. 3e** and **Supplementary Fig. 14**). This trend is due to the focusing process of polydispersed Au NCs by the enhanced etching ability of thiol ligands under basic condition³³. Nonetheless, the homogeneity of the final product in this condition is not optimal because of the existence of NCs larger than Au₂₅ NCs. The mass of the larger NC is matched with Au₃₈(MHA)₂₄ NCs which appears when the amount of NaBH₄ exceeds the 'stoichiometric amount'²⁶. In addition, the formation of Au NCs is rapidly initiated immediately following the NaBH₄ injection, as indicated by abrupt color change under both conditions (**Fig. 3c.f**).

(Page 11, line 217; in the revised manuscript)

(before)

On the contrary, in the synthesis of more homogeneous Au₂₅ NCs, 1:3 Au to MHA ratio and 15 seconds of first reduction time (**Fig. 4a-c**), prominent UV-Vis peaks of Au₂₅ NCs in the final product confirm the successful synthesis with enhanced homogeneity. (**Fig. 4a**) Notably, a peak located at approximately 520 nm, corresponding to smaller-sized NCs than Au₂₅ NCs⁴⁰, disappears after 1 hour of second reduction time. It indicates that homogeneous Au₂₅ NCs are formed via the growth of smaller NCs, also confirmed by mass spectra (**Fig. 4b**). This trend is also consistent in the synthesis with 1:2 Au to ligand ratio and 7 d of first reduction time, as evidenced by UV-Vis spectra and mass spectra, obtained during the second reduction (**Fig. 4d,e**).

(after)

On the contrary, in the synthesis of more homogeneous Au₂₅ NCs, 1:3 Au to MHA ratio and 15 seconds of first reduction time (**Fig. 4a-c**), prominent UV-Vis peaks of Au₂₅ NCs in the final product confirm the successful synthesis with enhanced homogeneity (**Fig. 4a**). Notably, a peak located at approximately 520 nm, presumably corresponding to intermediate Au₁₈(MHA)₁₄ NCs⁴²⁻⁴⁴, disappears after 1 hour of second reduction time. It indicates that homogeneous Au₂₅ NC synthesis proceeds with the consumption of smaller intermediate NCs^{25,26}, also confirmed by mass spectra (**Fig. 4b and Supplementary Fig. 15**). This trend is also consistent in the synthesis with 1:2 Au to ligand ratio and 7 d of first reduction time, as evidenced by UV-Vis spectra and mass spectra, obtained during the second reduction (**Fig. 4d,e**).

(Newly added references; in the manuscript)

43. Tlahuice, A. & Garzón, I. L. On the structure of the Au₁₈(SR)₁₄ cluster. *Phys. Chem. Chem. Phys.* **14**, 3737–3740 (2012).
44. Ghosh, A., Udayabhaskararao, T. & Pradeep, T. One-step route to luminescent Au₁₈SG₁₄ in the condensed phase and its closed shell molecular ions in the gas phase. *J. Phys. Chem. Lett.* **3**, 1997–2002 (2012).

(Syntheses; in the revised manuscript)

To separate $[\text{Au}(\text{MHA})_2]^-$ complexes from the Au NCs under the different conditions, the as-synthesized Au NCs were purified by ultrafiltration tube with a molecular weight cutoff (MWCO) of 3,000 Da. Pretreatment for the filtrated solution was not required for characterizations.

(Mass characterization; in the supplementary information)

The electrospray ionization mass spectra (ESI-MS) were obtained using ultrahigh-performance liquid chromatography-quadrupole time-of-flight mass spectrometer (SYNAPT G2-Si, Waters) centered at Korea Basic Science Institute (KBSI) Seoul center, Republic of Korea. The solution of samples was directly infused into ESI chamber without any treatment. Sample flow rate: 0.05 ml/min; The capillary voltage: 2.5 kV; temperature of source and desolvation: 120 °C; flow rates of cone and desolvation gas: 70 and 200 L/h, respectively; nebuliser pressure: 3 bar.

Thank you very much for many appropriate and valuable comments. We are sure that these comments improved significantly the quality of the manuscript.

Reviewer: 2

General Comments: The synthesis involving metal thiolate complexes is the key protocol to produce most of thiolate-protected metal nanoclusters, including the representative Au and Ag nanoclusters. Although the M-thiolate complexes produced from reaction between metal salts and thiols are direct precursor identified in the reduction reaction to final products, the whole synthetic systems are somehow too ‘dirty’ to find out the true relationship between distribution of precursor with the homogeneity of metal nanoclusters. In this paper, the authors successfully identified this key issue existing in the synthesis of thiolate-protected metal nanoclusters and used Au₂₅ nanocluster as the model system to find out the relationship between precursor and final product. Generally, I think this research is important, especially for discovery more methods to produce different kinds of metal nanocluster species. I am also exciting to see that the similar principle can be applied to Pt, Pd and Rh clusters. I support its publication after addressing the following a few concerns from me.

Response: We truly appreciate the reviewer for the encouraging comments. In accordance with the reviewer’s comments, we have tried to answer the questions in point-by-point manner.

Comment 1: For small Au-thiolate complexes and final products, maybe ESI-MS is a ‘softer’ method to get more precise information on their distribution and more sensitive than MOLDI-MS. Have you tried it and get some data?

Response 1: We appreciate the reviewer’s valuable questions. As the reviewer points out, we tried to obtain mass data of metal NCs via ESI-MS. Though we were not able to characterize the entire NCs by ESI-MS, we acquired some data of metal NCs for certain synthetic conditions. To begin with Au NCs, we acquired ESI-MS data of Au₂₅(MHA)₁₈ NCs under the condition of Au : MHA = 1:2, 15 seconds of the first reduction time, and 3 hr of the second reduction time in **Fig. 1,3 (Fig. R2.1)**. Though the spectrum is considerably noisy, the isotopic mass spectrum for Au₂₅(MHA)₁₈ NC with multiple charge were well identified. Interestingly, the mass spectrum for [Au₂₅(MHA)₁₉]⁰ NC, which is known to be produced by oxidative etching of [Au₂₅(MHA)₁₈]⁻ NC [*Nat. Commun.* **11**, 5498 (2020)], was also obtained. However, the mass spectrum for Au₃₈(MHA)₂₄ NCs, which were synthesized by the more amount of NaBH₄ than the ‘stoichiometric amount’ [*J. Am. Chem. Soc.* **140**, 11370–11377 (2018)] and identified by MALDI-TOF mass spectra as in **Fig. 1b** and **Fig. 3e**, was not identified by ESI-MS. Moreover, we tried to acquire ESI mass spectra of various metal NCs in **Fig. 5** as shown in **Fig. R2.2**. Some metal NCs which have not been reported previously were detected, but considering the MALDI-TOF mass spectra of those metal NCs in **Supplementary Fig. 23~25** in the revised supplementary information, the information of the detected ESI mass spectra of the metal NCs are very partial. For our synthetic systems in this study, where several sorts of metal NCs coexist in a single batch without delicate purification, MALDI-TOF MS is seemingly more suitable than ESI-MS to detect various species. However, ESI-MS was effective for obtaining mass information of some specific Au-MHA complexes, which will be articulated in **Response**

4.

Fig. R2.1 (For reviewer only). **a**, Broad ESI mass spectrum of Au NCs synthesized under the condition of Au : MHA = 1:2, 15 seconds of first reduction time, and 3hr of second reduction time. **b-d**, Zoom-in ESI mass spectrum of 1,2,3 labeled in **a**. The black-colored mass spectra in **b-d** are the calculated one of $[\text{Au}_{25}(\text{MHA})_{18}-3\text{H}]^{4+}$, $[\text{Au}_{25}(\text{MHA})_{18}-2\text{H}]^{3-}$, and $[\text{Au}_{25}(\text{MHA})_{19}-3\text{H}]^{3-}$, respectively. The color of the sample data is corresponding to that in **Fig. 1,3**.

Fig. R2.2 (For reviewer only). **a,c,e**, Broad ESI mass spectrum of Pt, Pd, and Rh NCs synthesized under the condition of **Fig. 5**, respectively. **b,d,f** Zoom-in ESI mass spectrum of metal NCs labeled in **a,c,e**, respectively. The black-colored mass spectra in **b,d,f** are the calculated one of $[\text{Pt}_7(\text{MHA})_5-2\text{H}+\text{Na}]^{2-}$, $[\text{Pd}_{14}(\text{MHA})_{10}-2\text{H}+\text{Na}]^{2-}$, and $[\text{Rh}_8(\text{MHA})_6-2\text{H}+\text{Na}]^{2-}$, respectively. The colors of the sample data are corresponding to those in **Fig. 5**.

(Page 10, line 202; in the revised manuscript)

(before)

This trend is due to the focusing process of polydispersed Au NCs by the enhanced etching ability of thiol ligands under basic condition³³, which is also revealed in the mass spectra of Au NCs (**Fig. 3e**). Nonetheless, the homogeneity of the final product in this condition is not optimal because of the existence of NCs larger than Au₂₅ NCs. In addition, the formation of Au NCs is rapidly initiated immediately following the NaBH₄ injection, as indicated by abrupt color change under both conditions (**Fig. 3c.f**).

(after)

This trend is due to the focusing process of polydispersed Au NCs by the enhanced etching ability of thiol ligands under basic condition³³. Nonetheless, the homogeneity of the final product in this condition is not optimal because of the existence of NCs larger than Au₂₅ NCs. **The mass of the larger NC is matched with Au₃₈(MHA)₂₄ NCs which appears when the amount of NaBH₄ exceeds the 'stoichiometric amount'²⁶.** In addition, the formation of Au NCs is rapidly initiated immediately following the NaBH₄ injection, as indicated by abrupt color change under both conditions (**Fig. 3c.f**).

Comment 2: How did you prepare the samples for EXAFS. Did you dry the sample? The handling of samples may change the precursor as they are so sensitive to the chemical environment.

Response 2: We appreciate the reviewer's insightful questions. For the EXAFS characterization of Au-MHA complexes, we dissolved the washed and precipitated complexes by using very low amount of H₂O as shown in **Fig. R2.3**. Such concentrated solution was held in our home-built holders of which hole is enough for X-ray to be penetrated. The Au-MHA complexes were quite stable that their chemical status was preserved after the EXAFS measurement, as confirmed by the mass spectrum of *ex situ* samples. The EXAFS spectra of Au NCs were also acquired with the concentrated solution of samples. For helping readers readily understand the experimental procedure, we added a sentence in Supplementary Methods in the revised supplementary information.

Fig. R2.3 (For reviewer only). **a**, (Upper) Digital photograph of highly concentrated aqueous Au-MHA complexes in **Fig. 2**. (Lower) Digital photograph of home-built holder for EXAFS experiment and schematic description of X-ray. **b**, Mass spectra of Au-MHA complexes before (black) and after (red) the XAS experiment.

(X-ray absorption spectroscopy (XAS); in the supplementary information)

X-ray absorption spectroscopic data of Au L3 edge were acquired from 10C beamline at Pohang Accelerator Laboratory (PAL), Republic of Korea. The energy of the storage ring electron beam was 2.5 GeV with a ring current of ~ 360 mA. The incident X-ray was monochromatized by a Si(111)/Si(311) double-crystal. Samples were concentrated in solution and contained in home-built container. The solution was measured by using transmittance mode. The spectrum is processed by using Athena and Artemis of Demeter package². The spectrum for a reference Au foil was also measured concurrently to calibrate each sample. **For XAS experiment, Au NCs and Au-MHA complexes were highly concentrated by dissolving the washed precipitates using small amount of H₂O. Such concentrated solution was held in our home-built holders of which hole is enough for X-ray to be penetrated.**

Comment 3: What are the approximate sizes of Pt, Pd and Rh clusters produced using MHA as ligand?

Response 3: We thank the reviewer for the question. The average and standard deviation in size of Pt, Pd, Rh metal NCs are shown in **Supplementary Fig. 23–25** in the revised supplementary information. Not only the size distributions of homogeneous metal NCs in **Fig. 5**, those of relatively polydisperse metal NCs synthesized under different conditions are also displayed. **Table R2.1** shows the size information of representative metal NCs in **Fig. 5**.

Table R2.1 (For reviewer only). Average and standard deviation in size of Pt, Pd, and Rh NCs

in Fig. 5.

	Pt NCs	Pd NCs	Rh NCs
Average size (nm)	0.96	1.26	1.05
Standard deviation (nm)	0.12	0.21	0.14

Comment 4: Have you observed the AuMHA₂ complex in your system? It is a major species involved in the growth process of Au nanoclusters.

Response 4: We appreciate the important questions from the reviewer. We are aware that the [Au(SR)₂]⁻ complex is a byproduct in the stoichiometric reaction from Au-SR complex to Au NC [*J. Am. Chem. Soc.* **140**, 11370–11377 (2018)]. Instead of washing the as-synthesized Au NCs by ‘precipitation and redispersion’ method using antisolvent, we tried to separate the solution containing byproducts from the as-synthesized Au NCs in Fig. 1 by using ultrafiltration tube and performed ESI-MS to detect the complex (Fig. R2.4). Interestingly, only the solution from the condition of Au : MHA = 1:1 and 15 seconds of first reduction time does not contain [Au(MHA)₂]⁻ complex, while the filtrated solution from the other three cases contains the complex (Fig. R2.4b). This result implies that Au NCs, under the condition of the low extent of the first reduction, are not synthesized by the stoichiometric reaction of Au-SR complexes. It strengthens our findings that low extent of Au-SR complex formation makes the remaining Au salts as major precursors to Au NCs, leading to the uncontrolled synthesis of Au NCs and hamper their homogeneity. We added sentences regarding the [Au(MHA)₂]⁻ complexes in the revised manuscript. We also added Fig. R2.4, as Supplementary Fig. 13 in the revised supplementary information. Moreover, procedure for obtaining [Au(MHA)₂]⁻ complex was included in the Syntheses part in revised manuscript, and that for electrospray ionization mass spectra was added in the Supplementary Methods part in the revised supplementary information.

Fig. R2.4 (Supplementary Fig. 13 in revised SI). a, Digital photograph of separated solution of side product (bottom) and concentrated Au NC solutions in Fig. 1, Experimental and calculated (black) ESI mass spectra of $[\text{Au}(\text{MHA})_2]^-$ complex in the filtrated solution under the different conditions. All colors are corresponding to those in Fig. 1.

(Page 10, line 181; in the revised manuscript)

In contrast, the intensity of Au-S peak at 1.87 Å considerably decreases in the other two cases (Fig. 2f,g), implying that the use of Au-MHA complexes enhances the homogeneity of Au₂₅ NCs in the second reduction. In addition, the solution of Au NCs synthesized under the condition of Au : MHA = 1:1 and 15 seconds of first reduction time does not contain $[\text{Au}(\text{MHA})_2]^-$ complex, the well-known byproduct from the reduction reaction of Au-SR complexes to Au₂₅ NCs (Supplementary Fig. 13)²⁶. It infers that, when the extent of complex formation is low, Au NCs are synthesized from different synthetic route where the starting precursors are different from Au-SR complexes.

(Syntheses; in the revised manuscript)

To separate $[\text{Au}(\text{MHA})_2]^-$ complexes from the Au NCs under the different conditions, the as-synthesized Au NCs were purified by ultrafiltration tube with a molecular weight cutoff (MWCO) of 3,000 Da. Pretreatment for the filtrated solution was not required for characterizations.

(Mass characterization; in the supplementary information)

The electrospray ionization mass spectra (ESI-MS) were obtained using ultrahigh-performance liquid chromatography-quadrupole time-of-flight mass spectrometer (SYNAPT G2-Si, Waters)

centered at Korea Basic Science Institute (KBSI) Seoul center, Republic of Korea. The solution of samples was directly infused into ESI chamber without any treatment. Sample flow rate: 0.05 ml/min; The capillary voltage: 2.5 kV; temperature of source and desolvation: 120 °C; flow rates of cone and desolvation gas: 70 and 200 L/h, respectively; nebuliser pressure: 3 bar.

Thank you very much for many appropriate and valuable comments. We are certain that these comments improved the overall quality of the manuscript.

Reviewer: 3

General Comments: The authors have examined the detailed processes of metal nanocluster synthesis and concluded that the metal-ligand complexes play critical roles. However, this conclusion is well-known in the community, especially, from the work of Jianping Xie, et al. (e.g., their recent work <https://doi.org/10.1016/j.xcrp.2020.100206>). Therefore, I don't think the present work is particularly novel and exciting. In addition, researchers have now been able to routinely obtain molecularly pure gold nanoclusters and their single crystals, so the impact of this work is limited in terms of control the homogeneity of final products.

Response: We appreciate the reviewer for the critical perspective. We regret that we were unable to persuade the reviewer about the novelty of our findings in the initial manuscript. In this letter, we would like to clarify how our results deliver new insights in the field of nanoclusters. Detailed explanations follow.

1. There have been several previous reports studying the synthetic mechanism of metal NCs, mainly Au NCs, or suggesting methods to produce high-quality metal NCs in terms of homogeneity. We made a table which listed the previous studies about such themes and briefly explained their contribution to the society of nanocluster (**Table R3.1**).

Table R3.1 (For reviewer only). List of previous studies regarding the synthetic mechanism of metal NCs or methodology for synthesizing homogeneous NCs.

Corresponding author	Journal	Year	Volume	Page	NC	Content
J.Xie	ACS Nano	2012	6	7920	Au ₂₅ (Cys) ₁₈	First using mild reducing agent CO for synthesizing Au ₂₅
J.Xie	Chem. Mater.	2013	25	946	Au ₁₀₋₂₅ (SG) ₁₀₋₁₈	Controlling pH in CO reduction for desired NCs
J.Xie	Small	2013	9	2696	Au _{15,18} (SG) _{13,14}	Controlling pH and using mild reducing agent TBAB
J.Xie	Angew. Chem. Int. Ed.	2014	126	4711	Au ₂₅ (SR) ₁₈	Using high pH to enhance etching ability of SR during NaBH ₄ reduction
J.Xie	J. Am. Chem. Soc.	2014	136	10577	Au ₂₅ (p -MBA) ₁₈	Identifying intermediate Au NCs during Au ₂₅ synthesis via CO reduction
J.Xie	Chem. Commun.	2016	52	9522	Au _x (m -MBA) _y	Controlling SR to Au ratio to obtain different-sized Au NCs/NPs
J.Xie	J. Phys. Chem. C	2017	121	10743	Au ₂₅ (MHA) ₁₈	Moderately elevating temperature for synthesis of

						Au ₂₅ (MHA) ₁₈
J.Xie	J. Am. Chem. Soc.	2018	140	11370	Au ₂₅ (p -MBA) ₁₈	Establishing a stoichiometric equation for Au ₂₅ NC synthesis with NaBH ₄
J.Xie	Cell. Rep. Phys. Sci.	2020	1	100206	Au ₂₅ (p -MBA) ₁₈	Separating the stoichiometric equation of Au ₂₅ NC into eight steps
J.Xie	J. Phys. Chem. C	2021	125	4066	Au ₂₂ (SG) ₁₈	Controlling various parameters to increase yield of Au ₂₂ (SG) ₁₈ NC
R. Jin	J. Am. Chem. Soc.	2008	130	1138	Au ₂₅ (PET) ₁₈	Cooling Au-SR complexes to form single-sized aggregation
R. Jin	J. Mater. Chem.	2009	19	622	Au ₂₅ (SR) ₁₈	Using THF as a solvent for one-phase synthesis
R. Jin	Chem. Mater.	2011	23	2209	Au _{25,144} (PET) _{18,60}	Using methanol and high SR to Au ratio for desired Au NC
R. Jin	J. Am. Chem. Soc.	2011	133	9670	Au ₁₉ (PET) ₁₃	Using mild reducing agent and prolonged etching time
R. Jin	Nanoscale Res. Lett.	2012	7	1	Au _m (PET) _n	Controlling stirring speed and NaBH ₄ injection time
R. Kornberg	Nano Lett.	2016	16	3348	Au _x (m -MBA) _y	Controlling reaction time between Au and SR
J. Ying	Adv. Mater.	2020	32	1906063	Au _x (SR, SeR) _y	Two-phase synthesis of various and high-quality Au NCs

Among the numerous previous studies about Au NC synthesis we highlighted, we could identify three features for their contents. **i)** Most studies for the mechanism of Au NC synthesis have focused on the evolution process from Au-SR complexes to intermediate NCs and desired final products. **ii)** The strategies for the successful synthesis of Au NCs focus on the kinetics of reduction from Au-SR complexes to Au NCs. **iii)** There also exist a few studies controlling the reaction step before the reduction of the complexes to NCs, but they mainly suggest technical optimization for the size or quality control of Au NCs, rather than figuring out the fundamental mechanism behind the suggested strategy. Our results can be distinguished from the previous studies that we struggle to establish a general principle for the relationship between the status of Au-SR complexes and the homogeneity of Au NCs by delicately

controlling the complex formation process. This work not only encompasses the study for the evolution process from the complexes to NCs (Fig. 3,4 and Fig. R3.1,2), but also suggests the reason for the success or failure of synthesizing homogeneous Au NCs in terms of differences in Au-SR complexes (Fig. 2, Supplementary Fig. 8~12). Based on this perspective, we strongly believe that our study can be differentiated from the previous studies with regard to novelty and depth of research. We added Fig. R3.1, and Fig. R3.2 as Supplementary Fig. 15,14, respectively, to help readers further understand the detailed reaction mechanism. We added sentences and revised the previous expressions regarding the intermediate Au NCs in the revised manuscript.

Fig. R3.1 (Supplementary Fig. 15 in revised SI). a,b, Zoom-in mass spectra of intermediate Au NCs in Fig. 4a~c and Fig. 4d~f, respectively. Dashed lines indicate the intermediate Au NCs labeled with numbers. c, Table of the molecular formulas labeled NCs in a,b with the number of free valence electrons in parentheses.

Fig. R3.2 (Supplementary Fig. 14 in revised SI). (Left) Zoom-in mass spectra of intermediate Au NCs in Fig. 3d~f. Dashed lines indicate the intermediate Au NCs labeled with numbers. (Right) Table of the molecular formulas labeled NCs in mass spectra with the number of free valence electrons in parentheses.

(Page 10, line 202; in the revised manuscript)

(before)

This trend is due to the focusing process of polydispersed Au NCs by the enhanced etching ability of thiol ligands under basic condition, which is also revealed in the mass spectra of Au NCs (Fig. 3e). Nonetheless, the homogeneity of the final product in this condition is not optimal because of the existence of NCs larger than Au_{25} NCs. In addition, the formation of Au NCs is rapidly initiated immediately following the NaBH_4 injection, as indicated by abrupt color change under both conditions (Fig. 3c.f).

(after)

The intermediate Au NCs are concurrently formed with Au_{25} NCs after the initiation of second reduction²⁶, but their fraction decreases along with the reduction time (Fig. 3e and Supplementary Fig. 14). This trend is due to the focusing process of polydispersed Au NCs by the enhanced etching ability of thiol ligands under basic condition³³. Nonetheless, the homogeneity of the final product in this condition is not optimal because of the existence of NCs larger than Au_{25} NCs. The mass of the larger NC is matched with $\text{Au}_{38}(\text{MHA})_{24}$ NCs which appears when the amount of NaBH_4 exceeds the 'stoichiometric amount'²⁶. In addition, the formation of Au NCs is rapidly initiated immediately following the NaBH_4 injection, as indicated by abrupt color change under both conditions (Fig. 3c.f).

(Page 11, line 217; in the revised manuscript)

(before)

On the contrary, in the synthesis of more homogeneous Au₂₅ NCs, 1:3 Au to MHA ratio and 15 seconds of first reduction time (**Fig. 4a-c**), prominent UV-Vis peaks of Au₂₅ NCs in the final product confirm the successful synthesis with enhanced homogeneity. (**Fig. 4a**) Notably, a peak located at approximately 520 nm, corresponding to smaller-sized NCs than Au₂₅ NCs⁴⁰, disappears after 1 hour of second reduction time. It indicates that homogeneous Au₂₅ NCs are formed via the growth of smaller NCs, also confirmed by mass spectra (**Fig. 4b**). This trend is also consistent in the synthesis with 1:2 Au to ligand ratio and 7 d of first reduction time, as evidenced by UV-Vis spectra and mass spectra, obtained during the second reduction (**Fig. 4d,e**).

(after)

On the contrary, in the synthesis of more homogeneous Au₂₅ NCs, 1:3 Au to MHA ratio and 15 seconds of first reduction time (**Fig. 4a-c**), prominent UV-Vis peaks of Au₂₅ NCs in the final product confirm the successful synthesis with enhanced homogeneity (**Fig. 4a**). **Notably, a peak located at approximately 520 nm, presumably corresponding to intermediate Au₁₈(MHA)₁₄ NCs⁴²⁻⁴⁴, disappears after 1 hour of second reduction time. It indicates that homogeneous Au₂₅ NC synthesis proceeds with the consumption of smaller intermediate NCs^{25,26}, also confirmed by mass spectra (**Fig. 4b and Supplementary Fig. 15**).** This trend is also consistent in the synthesis with 1:2 Au to ligand ratio and 7 d of first reduction time, as evidenced by UV-Vis spectra and mass spectra, obtained during the second reduction (**Fig. 4d,e**).

2. We would also like to emphasize that our results point out the overlooked aspects in the synthesis of Au NCs. As far as we understand, most Au NC syntheses assume that thiol reduction completely removes the original Au salt precursors and maintains Au-SR complexes in uniform chemical state. However, our data have proved that the thiol reduction does not fully convert the initial HAuCl₄ precursor to Au-SR complexes (**Fig. 2b**). The remaining Au salts can act as a reactant to the reduction by NaBH₄ and lead to deteriorate purity of Au NCs (**Fig. 2d,e and Fig. 3**). In detail, participation of Au salts can even change the major reaction route for the synthesis of Au NC, considering the absence of [Au(MHA)₂]⁻ complex in the solution of Au NCs synthesized under the condition of Au : MHA = 1:1, 15 seconds of first reduction time (**Fig. R3.3**). The [Au(MHA)₂]⁻ complex is well-known byproduct from the stoichiometric reaction from Au-SR complex to Au NCs [*J. Am. Chem. Soc.* **140**, 11370–11377 (2018)]. It indicates that the low extent of Au-SR complex formation makes the synthetic route of Au NCs different from the previously known reaction pathway, which strengthens the novelty of our study. For help readers deepen the understanding of synthetic mechanism we suggest, we added schemes summarizing the different synthetic mechanism from Au-SR complexes to Au NCs in

Fig. 3,4 respectively, and some descriptions for these schemes in the revised manuscript. We also added some sentences regarding the $[\text{Au}(\text{MHA})_2]^-$ complexes in the revised manuscript and Fig. R3.3 as **Supplementary Fig. 13** in the revised supplementary information. Moreover, procedure for obtaining $[\text{Au}(\text{MHA})_2]^-$ complex was included in the **Syntheses** part in revised manuscript, and that for electrospray ionization mass spectra was added in the **Supplementary Methods** part in the revised supplementary information.

(Fig. 3; in the revised manuscript)

Fig. 3. Fast second reduction kinetics and less homogeneous Au NCs. a,d, UV-Vis absorption spectra, b,e, MALDI-TOF mass spectra in the linear mode, and c,f, Digital photographs of Au NCs during the second reduction using precursors obtained applying the 1:1, 15 s, and the 1:2, 15 s conditions, respectively. Dashed lines in UV-Vis absorption and mass spectra indicate the evidence of Au_{25} NCs. Arrows in a indicate the decreasing fraction of Au_{25} NCs with increasing second reduction time, while those in d indicate the increasing fraction of Au_{25} NCs with increasing second reduction time. Arrows in b,e indicate the

existence of Au NCs larger than Au₂₅. **g**, Reaction mechanism scheme when the degree of the first reduction is low.

(Fig. 4; in the revised manuscript)

Fig. 4. Slow second reduction kinetics and more homogeneous Au NCs. **a,d**, UV-Vis absorption spectra, **b,e**, MALDI-TOF mass spectra in the linear mode, and **c,f**, Digital photographs of Au NCs during the second reduction using precursors obtained applying 1:3, 15 s and 1:2, 7 d conditions, respectively. Dashed lines in UV-Vis absorption and mass spectra indicate the evidence of Au₂₅ NCs. Arrows in **a,d** indicate an increasing fraction of Au₂₅ with increasing second reduction time. Arrows in **b,e** indicate the existence of Au NCs smaller than Au₂₅. Evidence of Au NCs smaller than Au₂₅ also appears in UV-Vis absorption spectra (black asterisks). **g**, Reaction mechanism scheme when the degree of the first reduction is high.

Fig. R3.4 (Supplementary Fig. 13 in revised SI). a, Digital photograph of separated solution of side product (bottom) and concentrated Au NC solutions in Fig. 1, Experimental and calculated (black) ESI mass spectra of $[\text{Au}(\text{MHA})_2]^-$ complex in the filtrated solution under the different conditions. All colors are corresponding to those in Fig. 1.

(Page 11, line 212; in the revised manuscript)

In summary, a schematic of the synthetic mechanism of less homogeneous Au NCs is proposed in Fig. 3g. Low degree of first reduction yields a small fraction of Au-MHA complexes, and the remaining Au salts majorly participate in the formation of Au NCs. The high reactivity of Au salts and the Au-MHA complexes, with relatively low Au-S coordination, induces fast second reduction kinetics and results in the synthesis of Au NCs with reduced homogeneity.

(Page 11, line 227; in the revised manuscript)

To summarize the synthetic mechanism as displayed in Fig. 4g, high degree of the first reduction consumes most of the Au salts, producing Au-MHA complexes with high Au-S coordination as the major species. The stability of these complexes enables the controlled synthesis of homogeneous Au₂₅ NCs driven by slow second reduction kinetics.

(Page 10, line 181; in the revised manuscript)

In contrast, the intensity of Au-S peak at 1.87 Å considerably decreases in the other two cases (Fig. 2f,g), implying that the use of Au-MHA complexes enhances the homogeneity of Au₂₅ NCs in the second reduction. In addition, the solution of Au NCs synthesized under the condition of Au : MHA = 1:1 and 15 seconds of first reduction time does not contain $[\text{Au}(\text{MHA})_2]^-$ complex, the well-known byproduct from the reduction reaction of Au-SR complexes to Au₂₅ NCs (Supplementary Fig. 13)²⁶. It infers that, when the extent of complex

formation is low, Au NCs are synthesized from different synthetic route where the starting precursors are different from Au-SR complexes.

(Syntheses; in the revised manuscript)

To separate $[\text{Au}(\text{MHA})_2]^-$ complexes from the Au NCs under the different conditions, the as-synthesized Au NCs were purified by ultrafiltration tube with a molecular weight cutoff (MWCO) of 3,000 Da. Pretreatment for the filtrated solution was not required for characterizations.

(Mass characterization; in the supplementary information)

The electrospray ionization mass spectra (ESI-MS) were obtained using ultrahigh-performance liquid chromatography-quadrupole time-of-flight mass spectrometer (SYNAPT G2-Si, Waters) centered at Korea Basic Science Institute (KBSI) Seoul center, Republic of Korea. The solution of samples was directly infused into ESI chamber without any treatment. Sample flow rate: 0.05 ml/min; The capillary voltage: 2.5 kV; temperature of source and desolvation: 120 °C; flow rates of cone and desolvation gas: 70 and 200 L/h, respectively; nebuliser pressure: 3 bar.

3. Moreover, the reviewer concerns the limitation of our study with regard to the homogeneity of finally produced Au NCs. We would like to mention that our research includes synthesis of both monodisperse and polydisperse Au NCs by high and low extent of complex formation. Among such Au NCs, those synthesized from the condition of Au : MHA = 1:3, 15 seconds of first reduction time in **Fig. 1** are molecularly pure enough to be compared with the fine syntheses in previous studies. To verify the molecular-level purity of synthesized $\text{Au}_{25}(\text{MHA})_{18}$ NCs under the condition, we checked the optical density of those NCs in **Fig. 1a** at 670 nm as 0.057. Considering that the solution was five times diluted during UV-Vis absorbance and the molar absorption coefficient of $\text{Au}_{25}(\text{SR})_{18}$ NCs at 670 nm is $8.8 \times 10^3 \text{ M}^{-1} \cdot \text{cm}^{-1}$ [*J. Am. Chem. Soc.* **127**, 5261–5270 (2005)], [*ACS Nano*, **6**, 7920-7927 (2012)], the yield of $\text{Au}_{25}(\text{SR})_{18}$ NCs under such condition is calculated to be 80.97 %. This value is almost identical to the ideal yield of Au₂₅ NCs from the stoichiometric equation [*J. Am. Chem. Soc.* **140**, 11370–11377 (2018)], which indicates that our results also reach the molecular purity like other previous reports.

Furthermore, the reviewer points out the absence of single crystal of pure gold nanoclusters. However, for the water-soluble metal NCs, it is widely accepted that the single crystal of those NCs for X-ray diffraction measurement are extremely difficult to obtain [*J. Am. Chem. Soc.* **134**, 16662–16670 (2012)]. It is presumably owing to the flexible aliphatic chain in thiol ligands of water-soluble metal NCs. Such ligands are significantly affected by electrostatic attraction / repulsion and dipolar interaction in water, which disturbs the driving forces for crystallization [*Nat. Commun.* **14**, 658 (2023)]. We attempted but failed to grow the

high-quality single crystals of $\text{Au}_{25}(\text{MHA})_{18}$ NCs from the condition of high extent of complex formation (**Fig. R3.4**). Therefore, we want to state that the absence of single crystal of $\text{Au}_{25}(\text{MHA})_{18}$ NCs does not indicate that our study is limited in terms of controlling the homogeneity of final products. For emphasizing the homogeneity of Au NCs synthesized by the high extent of complex formation, we added one sentence and one supplementary note about the calculated purity of Au_{25} NCs in the revised manuscript and revised supplementary information, respectively. Moreover, for avoiding confusion, we revised the known value of Au_{25} absorbance from 690 nm to 670 nm in the revised manuscript.

Fig. R3.4 (For reviewer only). Digital photograph of crystallization of $\text{Au}_{25}(\text{MHA})_{18}$ NCs synthesized under the condition of Au : MHA = 1:3 and 15 seconds of first reduction time, which does not meet the quality to be measured by X-ray diffraction.

(Page 7, line 107; in the revised manuscript)

As the homogeneity of Au NCs decreases, distinct Au_{25} NCs absorption peaks, centered at approximately 400 nm, 450 nm, and 670 nm in UV-Vis spectrum (dashed lines in **Fig. 1a**)³³, become less prominent, while the uniform Au NCs achieve high purity of Au_{25} NCs as confirmed by their optical density at 670 nm (**Supplementary note 2**).

(Supplementary note 2; in the supplementary information)

Discussion on the verification of Au NC homogeneity via UV-Vis absorbance

The fraction of synthesized Au_{25} NCs in **Fig. 1** can be indirectly estimated by the optical density at 670 nm (OD_{670}) of sample. We checked the OD_{670} of homogenous Au NCs synthesized under the conditions of Au : MHA = 1:3, 15 s at 670 nm as 0.057. The solution was five times diluted during UV-Vis absorbance. Considering the initial concentration of Au is 1 mM and the molar absorption coefficient of $\text{Au}_{25}(\text{SR})_{18}$ NCs at 670 nm is $(8.8 * 10^3 \text{ M}^{-1} \cdot \text{cm}^{-1})^{11,12}$, the yield of $\text{Au}_{25}(\text{SR})_{18}$ NCs under such conditions can be calculated as below:

Yield of Au₂₅ NCs

= {(OD₆₇₀ / Molar absorption coefficient) * (the number of Au in Au₂₅ NCs) / (initial concentration of Au)} * 100 %)

= {(0.057 * 5 cm⁻¹ / 8.8 * 10³ M⁻¹ · cm⁻¹) * 25 / 1 mM} * 100 % = 80.97 %

The value of 80.97 % is almost identical to the ideal yield of Au₂₅ NCs from the stoichiometric equation (79 ± 2 %) ¹³, which indicates the molecular-level purity of Au₂₅ NCs produced from the condition.

4. Most importantly, this work find its importance in that our mechanistic finding can be universally applied to finding optimal conditions for synthesizing uniform metal NCs composed of various types of metal including Ag, Pt, Pd, and Rh as Fig. 5. Au₂₅ NC synthesis in our study is used as a model system to interrogate the relationship between the extent of metal-thiolate complex formation and homogeneity of NCs. Our thorough study from the model system can provide general guidelines to synthesize metal NCs composed of non-coinage metals as well as well-known atomically precise Au or Ag NCs. Indeed, as far as we understand, there exist very few reports on synthesis of thiolate-protected Pt nanoclusters [*J. Colloid Interface Sci.* **423**, 123–128 (2014)], Pd nanoclusters [*Angew. Chem. Int. Ed.* **61**, e202208751 (2022)], and none of Rh nanoclusters until nowadays compared to the numerous reports on nanoclusters made of coinage metal (Au, Ag, Cu). We succeed in obtaining size data of those metal NCs via TEM and MALDI-TOF (**Fig. 5, Supplementary Fig. 23~25**), and acquiring the accurate composition of some metal NCs (**Fig. R3.5**). Moreover, the accurate mass spectra of some Pt-SR complexes and Rh-SR complexes have also never been reported in previous studies as far as we understand (**Supplementary Fig. 28,30**), which highlights the novelty of our results. Our results will provide general principles for the successful synthesis of thiolate-protected metal nanoclusters and further give significant impact on the field of nanoscience.

Fig. R3.5 (For reviewer only). **a,c,e**, Broad ESI mass spectrum of Pt, Pd, and Rh NCs synthesized under the condition of **Fig. 5**, respectively. **b,d,f** Zoom-in ESI mass spectrum of metal NCs labeled in **a,c,e**, respectively. The black-colored mass spectra in **b,d,f** are the calculated one of $[\text{Pt}_7(\text{MHA}-\text{H})_5-2\text{H}+\text{Na}]^{2-}$, $[\text{Pd}_{14}(\text{MHA}-\text{H})_{10}-2\text{H}+\text{Na}]^{2-}$, and $[\text{Rh}_8(\text{MHA}-\text{H})_6-2\text{H}+\text{Na}]^{2-}$, respectively. The colors of the sample data are corresponding to those in **Fig. 5**.

Thank you very much for the constructive comments. Your comments improved the overall quality of the manuscript.

Reviewer: 4

General Comments: This manuscript reported a very comprehensive study on the influence of the precursor metal-ligand complexes in the metal (Ag, Au) NC synthesis. By controlling the fraction of Au species showing different reactivity, the authors found that the extent of complex formation alters reduction kinetics and the homogeneity of Au NCs, and this trend can be universally applied to syntheses of Ag, Pt, Pd, and Rh NCs. I recommend the publication of this work on Nat. Comm. after considering follow revisions:

Response: We truly appreciate the reviewer for the valuable comments. In agreement with the reviewer's comments, we have tried to answer the questions in point-by-point manner.

Comment 1: The major point of this work, the homogeneity of metal-ligand complexes before the reduction affects homogeneity of product metal cluster, has been previously proposed by Zhu and Jin in their report of synthesis of Au₂₅ cluster [JACS, 2008]. The author should be aware of this.

Response 1: We appreciate the reviewer's valuable advice. We are aware of the reference which kept the temperature cooled down to unify the aggregation of Au-SR complexes, which leads to the synthesis of homogeneous Au₂₅(SR)₁₈ NCs [*J. Am. Chem. Soc.* **130**, 1138–1139 (2008)]. We also verified that the low temperature contributes to the successful synthesis of Au₂₅(MHA)₁₈ NCs (**Fig. R4.1**). In addition, the moderately elevated temperature also resulted in homogeneous synthesis of Au₂₅ NCs, which is known to be caused by the enhanced etching ability of thiol ligands [*J. Phys. Chem. C* **121**, 20, 10743–10751 (2017)]. The two conditions in **Fig. R4.1** do not dramatically enhance the homogeneity of Au NCs compared to the room temperature condition. Therefore, we assumed that, in the perspective of controlling Au-SR complexes, the two parameters we select in this work are more critical toward the extent of complex formation and homogeneity of Au NCs.

Furthermore, we obtained the DLS data of as-produced Au-MHA complexes in **Fig. 2** before NaBH₄ injection like the reference which reviewer selected (**Fig. R4.2**). Interestingly, all synthetic conditions create aggregation of Au-MHA complexes with similar-sized single distribution as the reference. It indicates that the homogeneous aggregation of Au-SR complexes is rather a necessary condition for the homogeneous synthesis of Au NCs. It supports the importance of our thorough study on how the amount and fraction of Au-SR complexes can affect the uniformity of synthesized Au NCs. We added one paragraph in revised manuscript and one supplementary note in supplementary information describing the effect of temperature. We added the two references regarding the effect of temperature on the Au NC synthesis. We added **Fig. R4.1** and **Fig. R4.2** as **supplementary Fig. 4, 11** respectively in the revised supplementary information. We also added descriptions for the DLS data of Au-SR complexes in the revised manuscript. The information of DLS equipment was also included in the **Supplementary Methods** part in the revised supplementary information.

Fig. R4.1 (Supplementary Fig. 4 in revised SI). **a**, UV-Vis spectra, and **b**, MALDI-TOF mass spectra (in the linear mode) of Au NCs synthesized under three conditions of different temperature. The ratio of Au precursor to MHA is fixed as 1:2, the first reduction time is fixed as 1 hour, and second reduction time under all conditions is fixed as 3 hours. Dashed lines indicate the existence of Au₂₅ NCs in both **a** and **b**. The data of 25°C is matched with those in **Supplementary Fig. 1**.

Fig. R4.2 (Supplementary Fig. 11 in revised SI). DLS spectrum of aggregated Au-MHA complexes under each conditions. The colors of samples are matched with the data in **Fig. 2**.

(Page 6, line 96; in the revised manuscript)

Other than the two parameters, there exist other parameters, reactant concentration or temperature, which can control the Au NC synthesis^{34,35}. We examined how they affect the first reduction or homogeneity of produced Au NCs (**Supplementary Fig. 3,4**). Concentration or temperature shows less effect on both the degree of first reduction and Au NC homogeneity than the first reduction time and the ratio of Au precursor to thiol ligands (**Supplementary**

note 1).

(Supplementary note 1; in the supplementary information)

Discussion on effects caused by changes in temperature

We also assessed the effect of temperature whether it could affect the degree of first reduction (**Supplementary Fig. 4**). To obtain the substantial effect of changing temperature, we kept the first reduction time as 1 hour. The condition of 1 hour of first reduction time and 1:2 Au to SR ratio in **Supplementary Fig. 1** was used as standard (25 °C). Au₂₅ NCs were also successfully produced regardless of changes in temperature. Interestingly, both low and high temperature resulted in slightly higher homogeneity of Au NCs than that produced from the original condition. The moderately elevated temperature is known to enhancing the rate of thiol etching, leading to the formation of homogeneous Au₂₅ NCs⁹. The low temperature is known to unifying the size and structure of Au-SR complexes which can help the successful synthesis of homogeneous Au₂₅ NCs¹⁰. Both reactant concentration and temperature can affect the synthetic kinetics and quality of Au₂₅ NCs, but these parameters are seemingly less critical toward the extent of first reduction.

(Page 8, line 144; in the revised manuscript)

The compositions of the detected complexes under each condition are seemingly similar. Additionally, those complexes in solution state form single-sized aggregation with similar hydrodynamic diameter investigated by dynamic light scattering (DLS), regardless of synthetic conditions (**Supplementary Fig. 10**)³⁴.

(Supplementary Methods; in the supplementary information)

Dynamic light scattering (DLS)

DLS experiments were performed by using Malvern Zetasizer Nano. The as-synthesized Au NCs and Au-MHA complexes were diluted with deionized water without washing process.

Comment 2: Discussions should be added about how the metal-ligand complexes were converted into the homogeneous metal NCs upon the NaBH₄ reduction. For instance, some mechanisms were recently reported for the formation of Au_n(SR)_m clusters from the specific Au-SR oligomers [The Journal of Physical Chemistry C 2022, 126 , 5980-5990; Journal of the American Chemical Society 2021, 143, 15224-15232; Small 2021, 17, 2000627].

Response 2: We appreciate the reviewer's valuable comments. The reviewer suggests very comprehensive and profound references about the detailed mechanism from the Au-SR

complexes to the intermediate Au NCs. Those can be applied to our results regarding the growth trajectory under the condition of high extent of complex formation in **Fig. 4**, where the Au-SR complexes are initially transformed into the intermediate clusters (**Fig. R4.3**). Such intermediate clusters' Au(0) core are nucleated from the interaction between $Au_n(MHA)_n$ complexes and BH_4^- ions, and the initial intermediate clusters grow to larger ones via atom-by-atom size evolution as the suggested references articulate [*J. Phys. Chem. C* **126**, 5980-5990 (2022)], [*J. Am. Chem. Soc.* **143**, 15224-15232 (2021)], [*Small* **17**, 2000627 (2021)]. We revised and added sentences about the emergence of intermediate clusters in **Fig. 4** in the revised manuscript. We also added the **Fig. R4.3** as **Supplementary Fig. 15** in the revised supplementary information.

Fig. R4.3 (Supplementary Fig. 15 in revised SI). **a,b**, Zoom-in mass spectra of intermediate Au NCs in **Fig. 4a~c** and **Fig. 4d~f**, respectively. Dashed lines indicate the intermediate Au NCs labeled with numbers. **c**, Table of the molecular formulas labeled NCs in **a,b** with the

number of free valence electrons in parentheses.

(Page 11, line 217; in the revised manuscript)

(before)

On the contrary, in the synthesis of more homogeneous Au₂₅ NCs, 1:3 Au to MHA ratio and 15 seconds of first reduction time (**Fig. 4a-c**), prominent UV-Vis peaks of Au₂₅ NCs in the final product confirm the successful synthesis with enhanced homogeneity. (**Fig. 4a**) Notably, a peak located at approximately 520 nm, corresponding to smaller-sized NCs than Au₂₅ NCs⁴⁰, disappears after 1 hour of second reduction time. It indicates that homogeneous Au₂₅ NCs are formed via the growth of smaller NCs, also confirmed by mass spectra (**Fig. 4b**). This trend is also consistent in the synthesis with 1:2 Au to ligand ratio and 7 d of first reduction time, as evidenced by UV-Vis spectra and mass spectra, obtained during the second reduction (**Fig. 4d,e**).

(after)

On the contrary, in the synthesis of more homogeneous Au₂₅ NCs, 1:3 Au to MHA ratio and 15 seconds of first reduction time (**Fig. 4a-c**), prominent UV-Vis peaks of Au₂₅ NCs in the final product confirm the successful synthesis with enhanced homogeneity (**Fig. 4a**). **Notably, a peak located at approximately 520 nm, presumably corresponding to intermediate Au₁₈(MHA)₁₄ NCs⁴²⁻⁴⁴, disappears after 1 hour of second reduction time. It indicates that homogeneous Au₂₅ NC synthesis proceeds with the consumption of smaller intermediate NCs^{25,26}, also confirmed by mass spectra (**Fig. 4b and Supplementary Fig. 15**). This trend is also consistent in the synthesis with 1:2 Au to ligand ratio and 7 d of first reduction time, as evidenced by UV-Vis spectra and mass spectra, obtained during the second reduction (**Fig. 4d,e**).**

(Page 11, line 227; in the revised manuscript)

To summarize **the synthetic mechanism as displayed in Fig. 4g**, high degree of the first reduction consumes most of the Au salts, producing Au-MHA complexes with high Au-S coordination as the major species. The stability of these complexes enables the controlled synthesis of homogeneous Au₂₅ NCs driven by slow second reduction kinetics. **During the slow second reduction, according to recent reports, the Au(0) core of intermediate Au NCs is initially nucleated from the interaction between high fraction of Au_n(MHA)_n complexes and BH₄⁻ ions⁴⁵, and those intermediate NCs grow to larger ones via atom-by-atom size evolution process^{8,46}.**

(Newly added references; in the manuscript)

45. Li, L., Wang, P. & Pei, Y. Mechanism of Nucleation of Gold(I) Thiolate Oligomers into Gold-Thiolate Nanoclusters. *J. Phys. Chem. C* **126**, 5980–5990 (2022).

46. Peng, J. *et al.* Exploration of Formation and Size-Evolution Pathways of Thiolate-Gold Nanoclusters in the CO-Directed $[\text{Au}_{25}(\text{SR})_{18}]^-$ Synthesis. *Small* **17**, 1–14 (2021).

Comment 3: In the final part of the work, the strategy was used to synthesize the Pt, Pd and Rh clusters. Did the exact molecular formula of the Pt-SR, Pd-SR and Rh-SR clusters determined by the MS? In addition, from the absorption curve, no molecular like feature absorption peak was observed for the Pt-SR, Pd-SR and Rh-SR clusters (Fig. 5)?

Response 3: We thank the important questions from the reviewers. Based on the MALDI-TOF mass spectra in **Supplementary Fig. 23~25**, we notice that various sort of metal NCs coexists even under the synthetic condition of **Fig. 5**. Though their size distribution is narrow as evidenced by Cs-TEM images, they are seemingly less focused than the atomically precise Au or Ag NCs. The detected Pt, Pd, and Rh NCs can be approximately assigned as $\text{Pt}_{7-13}(\text{MHA})_{4-8}$, $\text{Pd}_{14-21}(\text{MHA})_{10-18}$, and $\text{Rh}_{10-18}(\text{MHA})_{7-14}$, respectively. We also attempted ESI-MS for the non-coinage metal NCs in Fig. 5 and obtained the mass spectra of some specific NCs (**Fig. R4.4**). However, the information from ESI-MS is presumably very partial compared to that from MALDI-TOF MS. Moreover, the separation of those NCs was very tricky due to the continuous band in PAGE lane (**Supplementary Fig. 23~25**).

In addition, other than the absorption peak in **Supplementary Fig. 23~25**, we did not find any noticeable peaks in UV-Vis spectra in visible or NIR region. It is seemingly due to the convolution of several peaks from various sort of metal NCs as mentioned above. Observing the atomic structure of those type of metal NCs and assessing their electronic energy level via spectroscopy or calculation will be significantly interesting topics, but we think that it goes beyond the scope of the current research.

Fig. R4.4 (For reviewer only). a,c,e, Broad ESI mass spectrum of Pt, Pd, and Rh NCs synthesized under the condition of Fig. 5, respectively. b,d,f Zoom-in ESI mass spectrum of metal NCs labeled in a,c,e, respectively. The black-colored mass spectra in b,d,f are the calculated one of $[\text{Pt}_7(\text{MHA})_5-2\text{H}+\text{Na}]^{2-}$, $[\text{Pd}_{14}(\text{MHA})_{10}-2\text{H}+\text{Na}]^{2-}$, and $[\text{Rh}_8(\text{MHA})_6-2\text{H}+\text{Na}]^{2-}$, respectively. The colors of the sample data are corresponding to those in Fig. 5.

Thank you very much for many appropriate and valuable comments. I am sure that these comments improved significantly the quality of the manuscript.

Reviewer comments, second round

Reviewer #1 (Remarks to the Author):

This revised manuscript has addressed all reviewers' concerns. I think the manuscript is acceptable as it.

Reviewer #2 (Remarks to the Author):

The authors have addressed all the concerns, and I would suggest the acceptance of this paper.

Reviewer #4 (Remarks to the Author):

My questions were well addressed by the authors, I have no additional comments. The manuscript is acceptable in the present form.